# A Review of Emergency and Disaster Management in the Process of Healthcare Operation Management for Improving Hospital Surgical Intake Capacity

**Mohammad Heydari [1], Kin Keung Lai [2,3,*], Yanan Fan [4] and Xiaoyang Li [1]**

[1] Business College, Southwest University, Chongqing 400715, China
[2] International Business School, Shaanxi Normal University, Xi'an 710062, China
[3] Department of Industrial and Manufacturing Systems Engineering, Hong Kong University, Hong Kong
[4] Faculty of Earth Sciences and Environmental Management, University of Wrocław, 50-137 Wrocław, Poland
[*] Correspondence: mskklai@snnu.edu.cn

**Abstract:** To perform diagnosis and treatment, health systems, hospitals, and other patient care facilities require a wide range of supplies, from masks and gloves to catheters and implants. The *"healthcare supply chain/healthcare operation management"* refers to the stakeholders, systems, and processes required to move products from the manufacturer to the patient's bedside. The ultimate goal of the healthcare supply chain is to ensure that the right products, in the right quantities, are available in the right places at the right time to support patient care. Hospitals and the concept of a healthcare delivery system are practically synonymous. Surgical services, emergency and disaster services, and inpatient care are the three main types of services they offer. Outpatient clinics and facilities are also available at some hospitals, where patients can receive specialty consultations and surgical services. There will always be a need for inpatient care, regardless of how care models develop. The focus of this monograph was on recent OM work that models the dynamic, interrelated effects of demand-supply matching in the ED, OR, and inpatient units. Decisions about staffing and scheduling in these areas are frequently made independently by healthcare managers and clinicians. Then, as demand changes in real-time, clinicians and managers retaliate as best as they can to reallocate staffing to the areas that require it most at a particular moment in time in order to relieve patient flow bottlenecks. We, as OM researchers, must create models that help healthcare administrators enhance OR scheduling policies, ED demand forecasting, and medium- and short-term staffing plans that consider the interdependence of how demand develops.

**Keywords:** emergency management; surgical intake capacity; supply chain healthcare; healthcare operation management; operating rooms (ORs); operations management (OM); scheduling

**MSC:** 90B99

## 1. Introduction

Although there are numerous studies in the OM literature dealing with different aspects of the OR capacity supply–demand matching problem, the authors are not aware of systematic approaches for dealing with these issues being used by hospitals with which they have interacted. Often, ad-hoc approaches are used. (As we discussed in our earlier paper (How to Manage Red Alert in Emergency and Disaster Unit in the Hospital? Evidence From London), various hospitals may adopt various measures, such as (1) changing the nurses' scheduling to a sensitivity-based system (which rewards loyalty); (2) the emergency department implementing supply chain management tracking software (for immediate restocking of essential items); (3) ending double and triple shifts; (4) the D or D algorithm, which stipulates that you have ten minutes per patient to diagnose or discharge (this keeps patients moving through)). These approaches rely on descriptive data analysis and

relatively simple forecasting methods. A recent report addressed to the US President [1] mentions misaligned incentives as the primary reason for the lack of widespread use of systems engineering (loosely interpreted OM) techniques. Other reasons include a lack of expertise and the availability of relevant data. We agree with these assessments and provide an example in the remainder of this section that leverages our own experience. This example considers the problem of evaluating and adjusting OR block schedules. It demonstrates that OM-inspired ways of thinking can be brought to bear on important capacity decisions with the potential to improve operations significantly.

From an OM perspective, the problem of deciding how much block time to allocate to a service or an individual surgeon is a newsvendor problem. Staffed-OR utilization suffers if a hospital assigns too much block time. That can lead to higher staffing costs as well as the inability to offer block time to surgeons who do not have block allocations. In contrast, too little block time may cause surgeons to operate at other hospitals, leading to a loss of revenue. For employed physicians who do not have that option, too little OR time could increase the administrative burden associated with negotiating additional OR time on an ad hoc basis and lengthen wait times for their patients. This implies that an optimal block size balances the cost of having too little on the one hand and too much on the other hand, similar to the tradeoff in a newsvendor model. The solution to the newsvendor model is a percentile of the demand distribution, calculated at the critical ratio of the unit underage cost to the sum of the unit underage and unit overage costs.

Although similar to the newsvendor problem in a broad sense, the block size determination problem is different because of important practical realities. For example, the demand for block time is the sum of planned case lengths for surgeries that ideally should be completed on a particular day. If block time is not sufficient to accommodate all cases, the surgeon must either request additional time or choose a subset of cases to perform on that day. Thus, demand for block times occurs in discrete chunks, which must be fitted into finite staff shifts. Moreover, planned case lengths (planned case length is the amount of time for which the OR is booked for a particular surgical case) for identical surgeries may vary by surgeon, procedure, and patient. In fact, there exists research in the OM context that focuses on the problem of determining the optimal planned case lengths for each given sequence of surgeries. Another practical difficulty is how to estimate how much revenue would be lost if hospitals reduced block allocations to some surgeons and by how much do administrative costs increase when surgeons have to frequently negotiate for additional OR time. Finally, most hospitals have existing block allocations, and it is difficult to change them significantly without encountering pushback from surgeons. If surgeons move their cases to a different hospital, then that can result in revenue loss for the hospital.

For the reasons outlined above, we discuss a practical approach to help OR directors review and revise block allocations. We illustrate our ideas with the help of an example based on actual data from one of the three hospitals mentioned in (Section 2.4). In the OM literature, Olivares et al. [2] use structural estimating equations on planned surgical case length data to identify key drivers of such booking decisions. Using their approach and assuming that decision-makers make optimal decisions, one can estimate the implied shortage and overage costs underlying the choice of planned case lengths. Similar to that research, we use a newsvendor framework as the basis of our approach. However, in contrast to that paper, we are primarily interested in the appropriateness of block allocations to a group of surgeons who operate at the same hospital. The analysis must take into account the fact that block time utilization is affected by the extent to which cases arising on a particular day can be fitted into the assigned block.

The hospital, in this example, allocated time to surgeons in two modes—either as guaranteed blocks or as to-follow blocks. A surgeon with a to-follow block did not have a fixed start time for his or her cases. Instead, his or her cases would begin whenever the block surgeon would finish his or her OR day. Block surgeons typically start their day at 8 a.m. Blocks were either 4 or 8 h long, and the hospital staffed ORs for a standard 8 h shift. For all practical purposes, to-follow blocks could be viewed as blocks with a tentative start

time of 1 p.m. In practice, to-follow surgeons experienced significant variability in their actual first-case start times and in the number of cases they could perform. Quite apart from this, some surgeons used block-only OR time, some used both block and to-follow block times, and the remaining operated only in the to-follow blocks. More senior surgeons were more likely to have dedicated blocks.

As a first step, we calculated two metrics to classify surgeons. These were the demand–supply match and block efficiency scores. The former was the ratio of total surgery minutes to available block minutes over a 1-year data sample, and the latter was the ratio of total minutes that were scheduled in the block to the available block minutes (ABM). These definitions are provided below for convenience.

$$SDS = Supply - Demand\,Score$$
$$= \frac{TSM}{ABM} \tag{1}$$

$$BES = Block\,Efficiency\,Score$$
$$= \frac{TBM}{ABM} \tag{2}$$

Clearly, SDS ≥ BES. Upon calculating these metrics, we could identify efficient block allocations—i.e., those that had high block efficiency and well-matched demand and supply. Note these calculations assume that demand equals the scheduled surgery minutes, i.e., there is no lost demand. By in-block minutes, we mean all surgery minutes on the day that a provider has block time allocation, regardless of whether the surgery occurred within the block or not. The result of our analysis is shown in Figure 1. This representation allows a straightforward classification of surgeons according to their use of OR blocks. In the bottom left square, both SDS and BES are less than 50%. These surgeons could be candidates for the reduction in their assigned block size. Similarly, surgeons whose BES is less than 50% but SDS is greater than 50% could be suffering from inappropriate block configuration—either frequency, day-of-week, or length of each block.

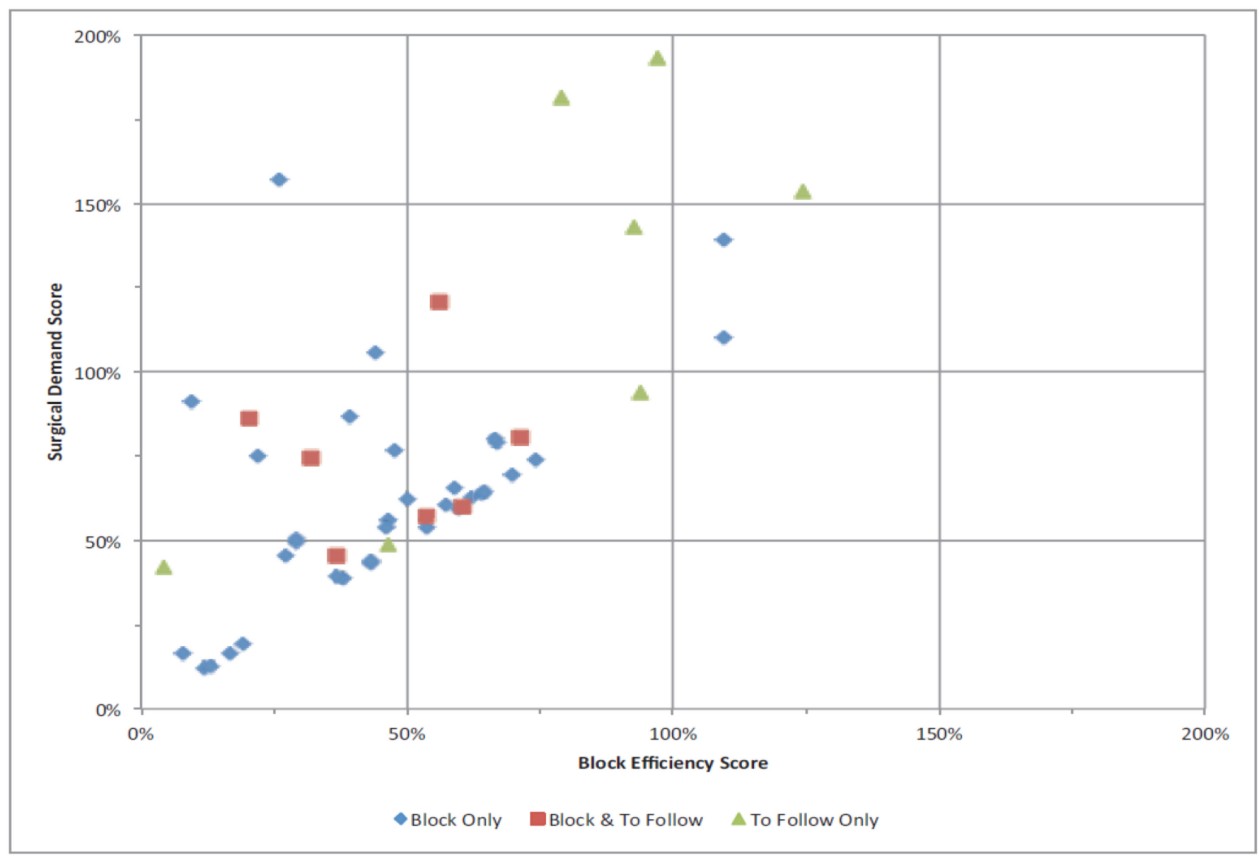

**Figure 1.** Demand–supply match and block efficiency scores of surgeons.

Surgeons whose SDS and BES both lie between 50% and 100% may be considered to have a reasonable allocation of blocks. Finally, those that have simultaneously high BES and high SDS (different hospitals may use different thresholds) may be deemed eligible for more block time. Once an agreement is reached regarding the criteria that will be used to evaluate block allocations, it is not too difficult to improve existing block allocations.

The problem of deciding how much block time to assign to different surgeons or surgical groups can be framed as a mechanism design problem. If this approach were followed, each surgeon would be suggested a menu of allotments, along with some incentives (e.g., gainsharing from reduced staffing costs), for maintaining high utilization of the assigned block time. The key challenge then would be to find a menu of block times and corresponding payment functions that would align the hospitals' and the surgeons' preferences. The hospital would pay informational rents because it would not know each surgeon's private cost of his or her effort to improve utilization.

Finally, in our research, we ask what difficulties hospital executives have in balancing supply and demand for medical services while preserving service quality and keeping prices low; to what extent and in what ways the OM literature has aided in the resolution of these issues; what the current practice trends are; and what extra opportunities and problems they present for operations management scholars. This study is the authors' attempt to answer these questions. Although service capacity might be measured in terms of the number of physical and HRs used, we chose to concentrate on the three primary types of services provided by hospitals.

In the present research, the notational scheme is used as follows. We use $I$, $j$, $m$, and $n$ to denote indices, e.g., the $i$th surgical case, the $j$th service line, the $n$th OR, and so on. Random variables are denoted by upper-case letters, and a random realization is denoted via the corresponding lower-case letter. All random variables are real-valued and non-negative.

*Terminology*

**Emergency management:** the organization and management of the resources and responsibilities for handling all humanitarian aspects of emergencies are known as emergency management, also known as emergency response or disaster management. The goal is to avert and lessen the negative consequences of all risks, including disasters.

**Disaster Management:** disasters can come in many forms. Industrial explosions or structural failures are examples of human-made disasters that are caused by mistakes made by humans. Earthquakes and droughts are examples of physical phenomena that result in natural disasters. Epidemics and armed conflicts are examples of complex disasters.

**Patient Intake:** patient intake is the procedure by which healthcare organizations gather vital information from both new and returning patients prior to their visits, including consent forms, insurance information, payments, and demographic, social, and clinical data.

**Operation Management Healthcare/Supply Chain Healthcare:** the term "*operations management*" in the context of healthcare refers to a facility's daily operations that have an impact on patient care and organizational objectives. Administrative, financial, and legal components make up these procedures most frequently.

**Ad-hoc Approach:** done only when needed for a specific purpose, without planning or preparation.

## 2. A Mathematical Framework for Current OM Approaches

Next, we present possible formulations for a subset of problems described in the last section, discuss and critique related papers, and highlight opportunities for future contribution. These problems represent a series of hierarchical decisions that OR stakeholders are likely to face. The discussed problems may be encountered by different stakeholders, e.g., some by OR directors and others by nursing directors, etc. There are several excellent reviews of OM literature on operating room capacity management. Some examples include Magerlein, Martin [3], Blake, Carter [4], Gupta [5], Gupta, Denton [6], Cardoen et al. [7],

Guerriero, Guido [8], and May et al. [9]. In addition, a comprehensive bibliography of operating room management research papers is maintained on the web by Dr. Franklin Dexter [10].

### 2.1. Number of ORs and Flexibility

At this stage of decision-making, the hospital is interested in choosing the number of operating rooms it should have. Such scenarios could arise either when demand patterns are perceived to have changed or when the hospital is undergoing renovation, or when a new hospital is being built. It is also important to decide how to equip each OR, or equivalently, which procedure types each OR should be able to handle. We focus on the first problem below.

For making long-term capacity decisions, it is often useful to think of a week as a time unit. Demand is aggregated across all procedure types and measured in minutes of OR time required on a weekly basis. Let $D_t$ denote the demand in period $t$. Demand distribution is assumed stationary and either empirically estimated or based on expert judgment. The decision variable is $n$, the number of ORs such that each OR provides $k\beta$ minutes of regular capacity, where $k$ is the theoretical capacity per period (e.g., $40 \times 60$ min per week for 8 h/day staff shifts) and $\beta \prec 1$ is the packing-efficiency factor. Some loss of capacity is inevitable because shift lengths and case lengths are discrete; cases cannot be split and must be fitted in discrete shift lengths. We call _ the packing efficiency of ORs. Different hospitals may implement different procedures to affect packing efficiency. For example, some may try to revisit OR schedules a few days before each surgery day in an attempt to improve OR utilization—see (Section 2.5: Improving Schedules). For this reason, we treat $\beta$ it as a discretionary variable in our calculations because management can affect it within some range.

In order to understand the range of possible values of the parameter $\beta$, it is relevant to consider both empirical evidence and the literature dealing with the efficiency of online bin-packing algorithms. The latter has been a topic of interest to the computer-science research community for some time. We discuss that literature first.

Assuming no additional constraints imposed by physician preferences, the performance of the best-known algorithm, which is due to Seiden [11], is no worse than 1.58889 times that of an optimal offline algorithm. Translated to our context, the value of $\beta$ upon utilizing the best scheduling algorithm and without considering surgeon preferences should be at least $1/1.58889 = 0.6294$ times the best solution under complete information. Note that even the best possible solution may not be able to fully utilize the available OR time, depending on the set of surgeries that need to be scheduled. Furthermore, there are certain unique features of the OR scheduling environment that may lower scheduling efficiency. For example, surgeons may prefer no downtime between their cases and to start their first case of the day as soon as the assigned OR opens in the morning. On the flip side, efforts to improve OR schedules described in (Section 2.5: Improving Schedules) can cause efficiency to be greater than the theoretical bound. Such issues are not considered in online bin-packing literature. A survey of this literature and key algorithms can be found in Coffman et al. [12].

Turning next to empirical evidence, we find that ORs in the US are reported to operate at the staffed-capacity utilization of 60–70% [13]. The two pieces of evidence are surprisingly close and suggest that one may expect a $\beta$ value in the range of 0.5 to 0.7 for a typical suite of ORs.

Each OR can provide additional capacity from the use of overtime. Given $n$, the cost of building ORs is already committed. Therefore, ongoing costs are those related to patient waiting and overtime. In this problem set, the hospital would choose $n$ first and then decide how much overtime to use in each period. The latter could be decided after observing demand. Let $q_t$ denote the number of minutes of OR time (regular plus overtime) used in period $t$. Then, patient wait time in period $t$, measured in terms of the backlog of surgery minutes, equals

$$W_t = (W_{t-1} + D_t - q_t)^+ \tag{3}$$

and the amount of overtime used equals

$$O_t = (q_t - nk\beta)^+ \tag{4}$$

It is customary to assume that patients' wait costs and overtime costs are linear and that there is a constraint on overtime availability. (As Childers and Maggard–Gibbons discussed in their research, costs of care in the ORs follow a linear pattern. Also, in our previous research we showed that ATD costs of materials for emergency relief are linear. We approved this issue in our previous research). Let $c_w$ and $c_o$ denote the unit waiting and overtime costs. Moreover, let $\bar{o}$ denote the maximum amount of overtime available in any period. Then, the hospital's per-period operating cost for each $n$ is $\lim_{m \to \infty} \frac{1}{m}[\sum_{t=1}^{m} E(W_t)c_w + E(O_t)c_o]$, subject to $O_t \leq \bar{o}$.

When costs are linear, it is easy to see that in each period, the hospital should either use as much available overtime as needed to bring the backlog to zero or none at all. That is, the optimal number of OR minutes used are either $q_t^* = \min\{W_{t-1} + D_t, nk\beta\}$, if $c_w \leq c_o$, or $q_t^* = \min\{W_{t-1} + D_t, nk\beta + \bar{o}\}$, otherwise. In each scenario, we need to estimate $E(W_t)$. In the former scenario, there is no overtime, but in the latter, we also need to estimate $E(O_t)$.

In each period, the maximum available OR time equals either $q = nk\beta\,(if\ c_w \leq c_o)$ or $q = nk\beta + \bar{o}\,(if\ c_w \succ c_o)$. Period-t demand is $D_t$, and all unmet demand is backordered. At the start of each period, the backlog $W_t$ is updated according to Equation (3) with the appropriate value of $q$. Extra minutes of capacity are wasted. When demand is discrete (measured in 1-min units), this allows us to write

$$P(W_t = k) = \sum_{x=1}^{k+q_t-1} P(D_t = k + q_t - x)P(W_{t-1} = x), \tag{5}$$
$$k = 1, 2, \ldots$$

$$P(W_t = 0) = \sum_{x=0}^{q_t} P(D_t \leq q_t - x)dP(W_{t-1} = x), \tag{6}$$

$$P(O_t = y) = \sum_{x=0}^{nk\beta+\bar{o}-1} P(D_t = nk\beta - x + y)P(W_{t-1} = x), \tag{7}$$
$$y = 1, \ldots, \bar{o} - 1,$$

and

$$P(O_t = \bar{o}) = \sum_{x=0}^{\infty} P(D_t = nk\beta + \bar{o} - x)P(W_{t-1} = x). \tag{8}$$

Suppose $D_t$ is stationary, and either $E(D) < nk\beta$ with $c_w < c_o$, or $E(D) < nk\beta + \bar{o}$ with $c_w \geq c_o$. Then, a stationary distribution of backlog and overtime usage exists. Even though the transition probabilities are relatively easy to write down, it is difficult to achieve closed-form expressions for the steady-state distributions of $W_t$ and $O_t$, with the result that papers that deal with this problem often rely on simulation. An added benefit of simulation is that the details of the surgery scheduling process can be modelled. In contrast, we used a packing-efficiency parameter to model this aspect of the decision problem. Goldman and Knappenberger [14] present an early example of a simulation model.

Lovejoy and Li [15] consider three different objectives—wait time to get on the surgeon's schedule, start-time reliability, and hospital profit. They develop a queueing–theoretic model in which the key daily decision that a decision-maker makes is the number of surgical cases that should be scheduled for each OR and the likelihood that a case will begin on time. The ideal time for each OR to be open each day is determined given these two parameters. All case lengths are assumed independent and identically distributed. Under these assumptions, Lovejoy and Li [15] provide a model formulation, some straightforward dominance results, and a variety of numerical examples.

The dominance results are of the following type: if, for a fixed choice of the number of scheduled cases, the hospital wants to increase the probability of an on-time start, then it must plan to staff the OR for a longer duration, and each procedure's allowance must also increase. Note that the model we presented earlier in this section is closely related, with the difference that in our model, hospitals choose the number of staffed ORs, and

overtime cases are fitted into available OR capacity. Like the model in Lovejoy and Li [15], our model also does not lead to closed-form analytical expressions. Lovejoy and Li [15] present sample calculations using data from a hospital.

Note that based on the framework we presented and those that can be found in previous works, it is possible to relatively quickly compute the number of ORs that would lead to the lowest overall cost. Unfortunately, these models do not result in an easy-to-use formula or a rule of thumb that will be easy to explain to OR managers. Instead, they provide computational techniques that can be brought to bear on each set of input data. Moreover, as input, all of these models require forecasts of future demand, which may pose additional challenges for hospitals that attempt to use these approaches.

We turn next to the question of determining the capabilities (dedicated equipment) for each OR. Such issues have been studied extensively in manufacturing and service industries under the banner of process flexibility; we refer readers to Buzacott and Mandelbaum [16] for a comprehensive and insightful review. Process flexibility stems from a system design that allows a firm to produce different types of products and/or services in response to changing demand, without incurring significant penalties in labor and materials costs. Flexible facilities (equipment) require greater up-front investment and may incur greater per-unit production costs, as opposed to dedicated facilities (equipment). However, flexibility provides a hedge against uncertain demand. Some recent articles address questions such as (1) 'how much flexibility is adequate?' (e.g., [17,18]), (2) 'how can manufacturing or service systems be designed for flexibility?' (e.g., [19,20]), and (3) 'what are the costs and benefits of utilizing flexible labor sources?' (e.g., [21]).

There is insufficient evidence in the healthcare context to assess the extent to which such ideas are useful when making OR capacity decisions. In addition, there are certain unique features of OR capacity allocation that prevent the direct application of previous results. For example, surgical equipment may be wheeled from one room to another; equipment needs to be sterilized between procedures, doctors routinely perform multiple cases in a day, and same-doctor cases must not overlap; different ORs may be staffed for different shift lengths, and so on. There is a need both for new analytical models that could help OR directors make decisions regarding the flexibility of ORs, as well as translational/implementation studies dealing with such issues.

### 2.2. Staffing Decisions

Suppose a hospital has historical data on scheduled start and end times of procedures, the physical number of ORs is fixed, and staff could work in either 8, 10, or 12 h shifts. Furthermore, the earliest shift start time is 7 a.m., the latest end time is 7 p.m., and time is measured in units of 15 min. How many staff shifts of each type should the hospital have as regular staff, and what should be their start and end times? We introduce a model next that can help answer the questions posed above. We consider each historical day of data for a particular weekday (e.g., Mondays), denoted by $w \in \Omega$, an equal-chance scenario. There is a total of $|\Omega|$ days of data. Inputs to our models include time index $t = 1, \cdots, 48$, where $t = 1$ refers to [7:00, 7:15] a.m., and $t = 48$ refers to [6:45, 7:00) p.m., $d_t^w$, the maximum number of procedures concurrently in progress at each time interval $t$ in scenario $w$, $c_i$, the unit cost, and $s_i$, the length of type-$i$ shift, where $i = 1, 2,$ and 3. In numerical examples presented later, we shall assume $s_1 = 8$, $s_2 = 10$, and $s_3 = 12$ h. The decision variables are $n_i$, the number of type-$i$ shifts used, and $y_{it}$, the number of type-$i$ shifts that start at time t. Both are integer-valued. We use $c_0$ to denote the cost per unit (i.e., per 15-min interval) of staffing an OR in overtime. Typically, overtime wages are 1.5 times the regular wages. The formulation shown below allows a user to specify $m_i$, the maximum number of available type-$i$ shifts.

Suppose the hospital has already picked $n_i$ and $y_{it}s$. Then, it can calculate the number of ORs concurrently staffed in overtime. We denote this quantity $z_t^w$ for time period $t$ and scenario $w$. The daily staffing cost can be calculated as $c_0(1/|\Omega|)\sum_w \sum_t z_t^w + \sum_i c_i n_i$, leading to the following problem formulation.

$$\min_{n_i, y_{it}} c_o (1/|\Omega|) \sum_w \sum_t z_t^w + \sum_i c_i n_i, \tag{9}$$

Subject to:

$$z_t^w \geq d_t^w - \sum_i \sum_{t-s_i \prec T \leq t} y_{iT}, \; \forall t, w \tag{10}$$

$$z_t^w \geq 0, \; \forall t, w \tag{11}$$

$$\sum_t y_{it} \leq n_i, \; \forall i \tag{12}$$

$$y_{it} \leq x_{it} M, \quad \forall i, t \tag{13}$$

$$\sum_t x_{it} \leq m_i, \quad \forall i \tag{14}$$

$$y_{it} \in I, \; \forall i, t \tag{15}$$

$$x_{it} \in \{0, 1\}, \quad \forall i, t \tag{16}$$

In Constraint (13), $M$ is a large integer, and in Constraint (15), $I$ is the set of integers. For our formulation, it would suffice to set $M$ slightly more than $\max_{t,w}\{d_t^w\}$.

The staffing-problem formulation described above has some limitations. For example, it does not consider the fact that staff schedules are typically chosen on a biweekly (pay-period) basis and governed by a variety of work rules and contractual agreements with specific employees. It may not be possible to have a significantly different number of shifts in each type-category across different days of the week, although some variation is possible because some staff work part-time. Similarly, there may be constraints on the degree to which shift start times could differ across employees. Finally, the above formulation attempts to find the best staffing under the assumption that historical patterns of demand for OR time will be repeated in the future. That may not hold, and in particular, the new staffing patterns may affect future demand patterns, making it necessary to periodically revisit the staffing problem. Alternatively, one may use a different (more general) characterization of uncertainty; see, for example, Bandi and Bertsimas [22].

Notwithstanding the shortcomings identified above, we used CPLEX to solve the above optimization problem using representative data from one of the hospitals. The results are reported in Table 1. For the purpose of this experiment, we assumed that shift start times must be selected in 15-min intervals and that the overtime cost was 1.5 times the regular wages per unit time. We also included one half-hour of non-productive time in each shift. This comes from the fact that staff is typically allowed at least two 15-min breaks during a workday. Thus, the number of productive hours in an 8 h shift is 7.5.

**Table 1.** Optimal staff schedules for a selected day-of-week.

| Selected Day of Week | Shift Types | Shift Start Times | | | | | | |
|---|---|---|---|---|---|---|---|---|
| | | 7:30 | 7:45 | 8:00 | 8:15 | 8:30 | 8: 45 | 9:00 |
| Current | 8 h | 10 | n.a. | n.a. | n.a. | n.a. | n.a. | n.a. |
| | 10 h | n.a. | n.a. | n.a. | n.a. | n.a. | n.a. | n.a. |
| | 12 h | 6 | n.a. | n.a. | n.a. | n.a. | n.a. | n.a. |
| Proposed | 8 h | 3 | n.a. | 3 | n.a. | n.a. | n.a. | 2 |
| | 10 h | n.a. | n.a. | 1 | n.a. | 1 | n.a. | n.a. |
| | 12 h | n.a. | 1 | n.a. | n.a. | n.a. | n.a. | n.a. |

Note. n.a. = not available or not applicable.

Table 1 shows both the actual and optimal shift schedules for an arbitrarily chosen weekday based on data obtained from one hospital. A similar pattern was observed for other weekdays. We observe that the original plan had ten 8 h shifts and six 12 h shifts. In contrast, the optimal arrangement has eight 8 h shifts, two 10 h shifts, and one 12 h shift. If we ignore the differences in shift lengths, this represents a saving of three shifts.

Whereas the result shown in Table 1 is merely an example, it shows the benefit of staggering shift start times and using multiple shift lengths at the same time. Multiple shift lengths do increase the difficulty of developing staffing plans. Similarly, staggered shift starts times require greater cooperation from staff, which may require the hospital to offer incentives to staff willing to work at non-standard shift start times. Therefore, some of the benefits observed in Table 1 come at the expense of greater administrative effort, as well as the cost of incentives. Still, these benefits are significant, and techniques similar to the one we presented above are worthy of further exploration by hospitals attempting to reduce OR staffing costs.

There is substantial OM literature on nurse staffing. Some of this literature deals with the optimal staffing levels; see, e.g., Miller et al. [23], Yankovic, Green [24], and Véricourt, Jennings [25]. Some other research deals with detailed scheduling of nurse shifts given nurse staffing levels and scheduling constraints; see, e.g., Lim et al. [26] and Bard, Purnomo [27,28]. These studies are not directly applicable to the issue of staffing ORs. An integrated approach, similar to the example we presented above, is likely to be more useful for OR staffing because daily caseloads, dictated by surgeons' booking patterns, exhibit significant variability. Finally, surgical case volumes can lead to significant variability in demand for inpatient beds, causing bottlenecks. In particular, McManus et al. [29] report that surgical case volumes often cause significant variability in demand for ICU beds.

### 2.3. Optimal Block Sizes

Suppose a hospital has determined an appropriate total block time for each service. The next key decision it needs to make is the size of each block. Ideally, the chosen block size should provide a good fit with the pattern of surgical case lengths that are scheduled by that service. We present a stylized formulation for this problem, which utilizes historical data concerning scheduled case lengths for different service lines. We recognize the limitations of our approach. For example, the historical data contains cases that were scheduled, not the true demand for different types of cases, and the future case mix may be different from the historical case mix [30–33].

Consider a particular recurring period during which the service performed $m$ surgeries; the number of blocks cannot exceed n; the start time, end time, and duration of the $i$th surgery are $s_i$, $e_i$, and $d_i$, respectively; and the maximum shift length is $t_{max}$. The problem is to decide the size of each block such that no block can exceed $t_{max}$, and there cannot be more than n blocks. An example would be a service line that receives 20 h of block time in a week. The hospital needs to decide on optimal block lengths, given that there cannot be more than three blocks. If the shift length may not exceed 12 h and blocks must be in multiples of 4 h, it could choose either two blocks, one for 12 h and the other for 8 h, or two 8 h blocks and one 4 h block. The optimal choice will depend on the duration of surgeries typically performed by the service.

Let $t_1$, $t_2$, ..., $t_q \geq 2m$, denote critical times, where critical times are obtained by taking the union of sorted start and end times of surgeries and convenient start and end times of blocks. For example, convenient start/end times may consist of clock time every hour and 30 min past the hour within the normal business day for the hospital. Some critical times may overlap. Let $(j, k)$ denote a combination of any two critical times $tj$ and $tk$. For $j < k$, we say that the block $(j, k)$ is feasible if duration $(t_k - t_j)$ meets the block feasibility requirements such as minimum and maximum block size specifications and the ability to contain at least one job. We preprocess data to identify sets $J$ and $I_i$, where $J$ is the set of feasible blocks, and $I_i$ contains surgeries that overlap surgery $i$. We allow an arbitrary specification of feasible blocks, e.g., we may want to restrict attention to blocks that start and end either on the hour or half past the hour.

The decision variables in the ensuing formulation are $x_{iu}$ and $y_{jk}^u$, where $u$ denotes the $u$th block $(1 \leq u \leq n)$, $i$ denotes the $i$th surgery $(1 \leq i \leq m)$, and $j$ and $k$ are critical points. Both $x_{iu}$ and $y_{jk}^u$ are binary in the sense that $x_{iu}$ is *1* if surgery $i$ is assigned to block $u$ and *0* otherwise. Similarly, $y_{ij}^u = 1$ if the $u$th block has critical points $j$ and $k$ as its start and

endpoints, and 0 otherwise. The problem of selecting block sizes could then be formulated as follows:

$$\max \sum_i d_i \sum_u x_{iu} \tag{17}$$

subject to:

$$y_{jk}^u = 0, \ \forall (j,k) \notin J \tag{18}$$

$$\sum_{j,k} y_{jk}^u \leq 1, \ \forall u = 1, \ldots, n \tag{19}$$

$$\sum_u \sum_{j,k} (t_k - t_j) y_{jk}^u \leq b \tag{20}$$

$$x_{ij} \leq \sum_{j,k} y_{jk}^u, \forall i, u \tag{21}$$

$$\sum_u x_{iu} \leq 1, \ \forall i \tag{22}$$

$$e_i x_{iu} \leq \sum_{j,k} t_k y_{jk}^u, \ \forall i, u \tag{23}$$

$$(t_{\max} - s_i) x_{iu} \leq \sum_{j,k} (t_{\max} - t_j) y_{jk}^u, \ \forall i, u \tag{24}$$

$$x_{iu} + x_{\updownarrow u} \leq 1, \ \forall i, \updownarrow \in I_i \tag{25}$$

$$y_{jk}^u, \ x_{iu} \in \{0,1\}, \ \forall i, u, j, k \tag{26}$$

The objective is to pick block sizes such that the amount of booked time that falls within the block is maximized. The first constraint ensures that a block is not picked if critical points $j$ and $k$ do not meet feasibility criteria. Constraint (19) ensures that each block is associated with at most a single pair of critical points. Constraint (20) ensures that the total block time does not exceed $b$. Constraints (21) and (22) prevent the solver from picking solutions, in which case $i$ is assigned to block $u$, but there is no block $u$, and cases in which surgery may be assigned to more than one block. Similarly, Constraints (23) and (24) guarantee that if case $i$ is assigned to block $u$, then the start and end times of case $i$ lie within the start and end times of the block. Constraint (25) prevents the solver from picking solutions in which conflicted cases are assigned to the same room, and Constraint (26) ensures that $x_{ij}$ and $y_{jk}^u$ are binary (0 or 1) variables.

The above formulation can be expanded to consider multiple days of data, where each day is treated as a scenario. For a particular service or surgeon, only those days would be considered on which he or she performs non-urgent surgeries. Suppose $w$ denotes the index of an arbitrary scenario, and $\Omega$ is the set of all scenarios. Each scenario is assumed to occur with equal probability, giving rise to the following deterministic equivalent program.

$$\max \sum_w \sum_i d_i^w \sum_u x_{iu}^w \tag{27}$$

subject to:

$$y_{jk}^{uw} = 0, \ \forall (j,k) \notin J \tag{28}$$

$$\sum_{j,k} y_{jk}^{uw} \leq 1, \ \forall u = 1, \ldots, n, w \in \Omega \tag{29}$$

$$\sum_u \sum_{j,k} (t_k - t_j) y_{jk}^{uw} \leq b, \ \forall w \in \Omega \tag{30}$$

$$y_{jk}^{uw1} = y_{jk}^{uw2}, \ \forall w1, w2 \in \Omega \tag{31}$$

$$x_{iu}^w \leq \sum_{j,k} y_{jk}^{uw}, \ \forall i, u \tag{32}$$

$$\sum_u x_{iu}^w \leq 1, \ \forall i \tag{33}$$

$$e_i x_{iu}^w \leq \sum_{j,k} t_k y_{jk}^{uw}, \ \forall i, u, w \tag{34}$$

$$(t_{\max} - s_i)x_{iu}^w \leq \sum_{j,k}(t_{\max} - t_j)y_{jk}^{uw}, \ \forall i, u, w \tag{35}$$

$$x_{iu}^w + x_{\updownarrow u}^w \leq 1, \ \forall i, \updownarrow \in I_i^w \tag{36}$$

$$y_{jk}^{uw}, \ x_{iu}^w \in \{0, 1\}, \ \forall i, u, j, k, w \tag{37}$$

In the above formulation, Constraint (31) guarantees that we do not anticipate realized surgery requests when deciding the block start and end times. For Constraint (36), we preprocess data to obtain a set $I_t^w$ $i$ for each day of operations.

The formulations shown above are *NP-hard* (if an algorithm for solving a problem can be used to solve any **NP-problem** (nondeterministic polynomial time) problem, then the problem is **NP-hard**. **NP-hard**, therefore, means "*at least as* **hard** *as any* **NP-problem**", although it might, in fact, be harder) because the knapsack problem is a special case of the above formulation. In general, such problems may be difficult to solve within a reasonable time. However, the practical size of such problems is not too large, and such problems can be solved with the help of general-purpose optimization software such as CPLEX. For example, block hours for a particular service line in a data sample belonging to one of the hospitals we described in (Section 2.4) did not exceed 24 per day, and the maximum number of blocks per service line was three.

Using data from that same hospital and solving the above formulation for a representative sample of surgical services, we obtain the results shown in Table 2. Observe that the block utilization could be improved by changing the block configuration while keeping the number of blocks and total block minutes fixed. The reason is that for services with multiple blocks per day, it helps to divide the block time unequally into one longer and one shorter block. That facilitates the scheduling of long and short cases more efficiently. In some cases, greater efficiency results from moving the block start times to match the time of day when surgeries typically begin.

**Table 2.** Optimizing block configuration—an example.

| Service | DoW | $|\Omega|$ | $n$ | B (min) | Initial Config | Revised Config | W (min) | A/W | Initial Unit | Revised Unit |
|---------|-----|-----------|-----|---------|----------------|----------------|---------|-----|--------------|--------------|
| A | M | 49 | 2 | 900 | 7:30–15:00<br>7:30–15:00 | 7:30–12:00<br>7:30–18:00 | 46,588 | 0.95 | 55 | 66 |
| A | T | 50 | 2 | 900 | 7:30–15:00<br>7:30–15:00 | 7:30–14:00<br>7:30–16:00 | 50,438 | 0.89 | 62 | 69 |
| A | W | 51 | 2 | 900 | 7:30–15:00<br>7:30–15:00 | 7:30–13:45<br>7:30–16:15 | 46,115 | 1.00 | 53 | 65 |
| A | TR | 50 | 2 | 1020 | 7:30–15:00<br>7:30–17:00 | 7:30–13:30<br>7:30–18:30 | 48,119 | 1.06 | 59 | 61 |
| A | F | 49 | 2 | 900 | 7:30–15:00<br>7:30–15:00 | 7:30–13:30<br>7:30–16:30 | 47,134 | 0.94 | 53 | 61 |
| B | F | 45 | 2 | 1020 | 7:30–16:00<br>8:30–17:00 | 7:30–15:00<br>7:30–17:00 | 28,620 | 1.60 | 49 | 51 |
| C | M | 27 | 2 | 720 | 9:30–13:30<br>9:00–17:00 | 9:00–13:30<br>10:00–17:30 | 11,876 | 1.64 | 40 | 45 |
| C | T | 34 | 2 | 720 | 9:00–17:00<br>11:00–15:00 | 9:00–17:00<br>10:30–14:30 | 14,946 | 1.64 | 40 | 50 |

DoW = day of the week; $|\Omega|$ = number of data days; W = total work in minutes; A/W = ratio of available minutes to work minutes.

To our knowledge, the block configuration issue has not been studied in the literature. It is a variant of the bin-sizing problem, which has been studied extensively for other application areas. We believe that models such as the one presented above can provide a basis for further discussion among OR management teams to find a block configuration that makes scheduling more convenient and improves block utilization.

### 2.4. OR Scheduling

The OR scheduling issue is the most widely studied problem in the OM literature. In a typical model setting, researchers assume that all of the decisions mentioned earlier in this paper have been made.

Furthermore, it is also assumed that the OR planner knows the complete set of surgical cases that need to be scheduled on a particular day in a particular OR, the distribution of surgical case lengths, and that each surgeon operates in a single OR only. The goal is to minimize the weighted sum of physician delay costs and costs related to the use of overtime via selecting case start times and the sequence in which surgical cases ought to be performed.

For the problem mentioned above, the following formulation is fashioned after the model represented in Denton, Gupta [34]. The problem consists of choosing case start times of n cases. The patients and the providers are punctual, and there are no cancellations or add-ons. Let $Z$ denote the vector of random surgery durations, the vector of scheduled start times, $W$ and $S$ the vectors of waiting and OR idle times for a given a and $Z$, d the length of the day, and $L$ the tardiness for a given a. Scheduled case lengths, which can be obtained from the knowledge of $a$, is denoted by $x$. In particular, $x_i = a_{i+1} - a_i$ for $i = 1, \cdots, n - 1$, and $a_1 = 0$. The waiting, idleness, and tardiness are then determined as follows:

$$W_i = (W_{i-1} + Z_{i-1} - x_{i-1})^+, \ i = 2, \ldots, n. \tag{38}$$

$$S_i = (-W_{i-1} - Z_{i-1} + x_{i-1})^+, \ i = 2, \ldots, n. \tag{39}$$

$$L = \left( W_n + Z_n - \sum_{i=1}^{n-1} x_i - d \right)^+. \tag{40}$$

Suppose $c^w$, $c^s$, and $c_{\updownarrow}$ denote per-unit costs of waiting, idling, and tardiness. Then, the operating room director's problem is

$$\min x \left\{ \sum_{i=1}^n c_i^w E|W_i| + \sum_{i=1}^n c_i^s E|S_i| + c_{\updownarrow} E|L| \right\}, \tag{41}$$

where the expectations are over $Z$. In practice, it is often difficult to estimate $c_w$. Moreover, costs are incurred by different stakeholders. The hospital, which pays the staffing costs, cares greatly about OR idle time and overtime, but it might not have an effect on physicians' compensation. Similar to this, patients are frequently asked to arrive early, and the waiting costs are largely covered by patients, if the same surgeon performs all procedures in an operating room on a given day. Among other stakeholders, anesthesiologists are concerned about the difference between anticipated and actual start times because it may have an impact on their compensation and show how their workdays can vary in length. Finally, a solution of (41) is insensitive to individual patients' wait times when $c_i^w = c^w$ for all $i$, which is a common assumption. In this sense, the formulation allocates total waiting time without considering fairness to different patients.

Papers dealing with the problem of choosing surgery start times fall into two categories. Both variants assume that all cases that need to be scheduled on a given day are known. It is assumed that actual durations can be sampled from an existing database of surgery durations where surgery durations are random, but their distributions are known [34–37], in the first case, and that they are unknown in the second case [38]. We point out certain features of the actual surgery booking process that are not modeled well by the above-mentioned approaches. First, surgeries are booked one at a time in many US hospitals. Surgeon offices call the hospital booking clerk to book cases as the need arises. Non-urgent cases are booked first, followed by urgent and emergent cases. Block surgeons have guaranteed allocations that allow them the ability to schedule cases on their OR day. Second, many surgeons perform multiple cases on their OR day, which may be performed in multiple ORs. Therefore, surgery bookings must avoid same-surgeon overlap. Such constraints are not considered in the problem formulations mentioned above. Third, scheduling papers only take into account one type of urgency; thus, they either entirely

concentrate on non-urgent cases or entirely concentrate on urgent/emergent cases. In articles that take into account both types, like [39], discrete surgery durations are not modeled. In other words, they presume that surgery scheduling is flexible.

Significant attention is paid in the literature to the estimation of the distribution of surgery durations. Many hospitals have invested in software that helps keep track of surgery bookings on each future day. This software also tracks planned and actual case lengths and predicts future case durations based on the surgeon, the procedure type, and patients' characteristics. Surgeons are given an opportunity to modify the scheduled case lengths based on their experience. As more hospitals adopt such systems, the estimation of surgery case lengths is likely to improve, although surgery durations will remain random. That is, it is unrealistic to expect either that all surgeries will start on time or that there will be no OR idle time and no staff overtime.

It is frequently challenging for OR managers to gain support from surgeons, even though a model-related solution can significantly improve the process for setting start times (see Denton, Gupta 2003 for details [34]). Successful surgeons frequently run their own operating room with 24/7 access. Although it is an expensive option for the healthcare system, it enables surgeons to use their time as efficiently as possible.

### 2.5. Improving Schedules

A typical scenario in many hospitals unfolds as follows. The OR management team examines the surgical schedule for one or two days in advance and attempts to manually adjust case start times to reduce the number of operating rooms that would need to be open at the same time. Costs related to hiring are reduced. A surgical schedule with scheduled start times and planned lengths of surgical cases is currently in place. The latter is based on a combination of values prompted by the scheduling software used by hospitals and input from the surgeons' offices. The management team only modifies the case start times and treats the surgical case lengths as fixed.

The mentioned OR rescheduling issue can be seen as a subset of the bin-packing issue, with bins standing in for staffed operating rooms and items or jobs for surgeries. The objective is to reduce the weighted sum of bins used (or, equivalently, the cost of staffed operating rooms), where the weight of a bin is proportional to its size, because hospitals may use more than one standard shift length. Because hospitals may employ staff with varying shift lengths and because surgeries performed by the same surgeon cannot overlap, this problem differs from other bin-packing problems studied in the literature.

Because the bin-packing problem is *NP-hard*, it is easy to check that the problem of improving OR reschedules is also *NP-hard.* In such instances, it makes sense to establish bounds, which can lead to an efficient implementation of the branch-and-bound approach. In a recent paper, Li et al. [40] developed such bounds and identified significant opportunities for reducing OR staffing costs by rescheduling [40]. In order to understand how rescheduling would impact surgeons' workdays, delays in surgery start times, and overtime usage, the authors also analyze the resulting OR schedules. We briefly summarize this paper next.

Li et al. [40] developed a framework to improve OR schedules consisting of three steps. In step one, they classify linked sequences of surgeries into chains. Additionally, they divide doctors into various groups based on the characteristics of the chains created by their operations. Step two uses surgeon classification to assign surgeries to ORs in a specific order, producing a lower bound and assisting in step three's recovery of a feasible solution that is no greater than (3/2) of the lower bound. By using their algorithm on data from three hospitals, the authors also evaluate the effectiveness of their solution. This reveals a number of things.

First, rescheduling flattens the peak number of concurrently staffed ORs and evenly distributes surgeries throughout the day, as is expected. Additionally, according to Li et al.'s research [40], efficiency is higher when a hospital is able to schedule a few long shifts due to that results in more effective packing of surgical cases. Second, efficiency improvements

come at the expense of more staff working OT, more downtime for surgeons, and longer surgeon stays in the hospital. Hospitals may be able to win the cooperation of surgeons by implementing a suitable gainsharing plan, according to Li et al. [40], who analyzed the effects of rescheduling on medical professionals and found significant cost savings from increased efficiency.

*2.6. Major Incident Situation Happened; How Do CCUS and Specialist Staff Act?*

Inpatient care is required for between 5 and 15 percent of patients who arrive at the hospital after a bombing or other mass casualty terrorist incident [30–33,41]. Less information is available regarding the demands made on critical care resources during other incidents, such as large fires or natural disasters [42]. However, staff at the Charity Hospital in New Orleans had to deal with increased demand after Hurricane Katrina without having the chance to leave for several days [43].

According to recommendations made by the US Taskforce for Mass Critical Care in 2007 [30,44], CCU plans could represent emergency mass critical care at three times the current capacity for up to ten days. A major incident necessitates good planning and organization in order to release beds or expand the source because CCU beds and specialists/staff are a limited resource that are typically fully utilized. It is also critical to establish early connections with various hospitals in order to transfer more stable patients in the safest manner possible, as critical care sources will frequently represent the main limiting factor when encountering large numbers of casualties. Based on this, critical care managers/leaders must be involved in planning [31] for mass casualties in the healthcare communities.

The CHEST Task Force for Mass Critical Care has suggested different levels of capacity expansion needs via different levels of casualty [45–47].

Critical care sources require enhancement as follows:

(a) Capacity of a conventional response at least 20 percent greater than the baseline incentive care unit maximum;

(b) Crisis response is able to expand via at least 200% above baseline incentive care unit max capacity via regional, local, national, and international agencies;

(c) Ability to expand quickly in the event of an emergency by at least 100% above baseline incentive care unit maximum capacity by utilizing local and regional resources.

The minimum requirements for critical care offered via the EMCC taskforce [43] are:

- Vasopressor administration
- Mechanical ventilation
- Sedation and analgesia
- If recommended by the hospital or a region, the best therapeutics and interventions, such as renal replacement therapy and nutrition for patients who cannot eat by mouth
- IV fluid resuscitation
- Antidote or antimicrobial administration for special disease processes, if applicable
- Algorithms to decrease adverse consequences of critical care and critical illness.

It is suggested that a tiered response be implemented, allowing for the consideration of progressively more high-risk management techniques as the incident's impact grows. In the end, patients would also be triaged for availability of limited critical care sources.

Predicting and managing such an incident requires:

- Training and education of specialists and staff
- A degree of equipment stockpiling or recognition of substitute resources (e.g., use of anesthetic ventilators to supply ventilatory support and NIV machines)
- Recognition of specialists and staff via transferable skills like recovery nurses, respiratory nurses, and previous critical care nurses;

When there is no way to dismiss patients who are still coming in, hospitals go into red alert. In this situation, where there are more people coming in than are leaving and there is nowhere for them to go and no spare capacity (beds), hospitals reach the point

where staff may genuinely wonder if they can treat all patients. A red alert indicates that there is a serious threat to the patients' safety. All non-emergency surgeries should be postponed in this situation, and the need emerges to look for free beds and start releasing patients from their beds until the flow improves. All of the wards become under pressure to discharge patients under these circumstances, and patients must be followed as they leave the hospital.

The emergency admission system becomes dysfunctional when all wards are full and there are no beds available in any ward. In such circumstances, hospital operations can break down entirely if one crucial person falls. Staff experience blockage as a result of the flow ceasing, with repeated returns to the front door. When a hospital is on alert, all wards are under pressure to discharge patients [31].

## 3. A Mathematical Framework for Emergency Departments (EDs)

Emergency departments (EDs), also called accident and emergency (A&E) departments in some countries, provide immediate treatment to accident and trauma patients with life-threatening conditions. They also provide treatment to patients who present with urgent-care needs. EDs are staffed around-the-clock, every day of the week, by highly trained medical professionals. Modern EDs possess advanced diagnostic and treatment capabilities.

Patients do not require a prior appointment to be treated in the ED. Demand for ED services is affected by patients' perceptions of how urgently they need care. Patients' decisions may also be influenced by family members, ambulance crews, primary- and urgent-care physicians, nursing-home staff, police officers, and national emergency telephone operators (for example, 911 operators in the US). (As the first point of contact for both a patient and an incident, an ambulance service gives the health system early notice of changes in the general operating environment).

Many countries have experienced significant increases in ED demand, e.g., annual increases of 5.3% in Australia and 5.9% in the United Kingdom and Switzerland have been reported over time periods ranging from 3 to 7 years. In the US, there has been a 41% increase in ED visits between 1995 and 2011 [48]. At the same time, the number of EDs in non-rural areas declined from 2446 to 1779 from 1990 to 2009 [49]. These trends are at least in part responsible for long ED wait times and other undesirable outcomes [50–58] and naturally lead to a series of questions regarding the reasons for the increased demand, the performance metrics that the public and the policymakers (do and ought to) care about, and what OM researchers can do to address the ED wait-time issue.

Our primary focus is on the third question posed above. Answers to the first two questions, discussed in (Section 3.1), provide the context for current and future OM researchers. Specifically, the bulk of this section concerns the following topics:

1. What factors can most effectively sculpt (even out) the demand for ED services?
2. How can the ED service processes be improved? Improvement is measured with respect to certain performance metrics that we discuss in (Section 3.1).
3. How can hospital managers ensure that there is an adequate supply of downstream beds for ED patients who may need additional hospital services?

For each of the questions posed above, we provide an overview of practitioner challenges, the current state of practice, and current OM approaches for addressing them. We conclude with opportunities for future OM research on ED demand–supply matching.

### 3.1. ED Demand

ED demand exhibits both predictable hourly and day-of-week patterns, as well as unpredictable (stochastic) variability in inter-arrival time [59]. Patients arrive in EDs either as walk-ins or in ambulances. Hospitals have no forewarning of walk-in arrivals, but emergency medical technicians do provide some advance information about patients who arrive in an ambulance. This information helps hospitals ensure ED bed availability when the ambulance arrives.

*Possible Reasons for Increasing Demand.* There are a variety of reasons that explain the sustained increase in ED demand over the years. Understanding these causes is important because they provide insight into how demand may be smoothed to make ED operations more efficient and effective. The most frequently cited reason for increased ED demand is the lack of timely access to primary care [60–63]. In 2010, 28% of primary care provider visits in the US took place in EDs [64]. Other reasons include an ageing population, higher public expectations for more convenient healthcare, the public accessing EDs rather than more appropriate lower levels of care, and an increase in the reliance on professional healthcare rather than self-care or other social structures for care [51,65].

The US Congress passed the EMTALA in 1986, which mandated that any hospital receiving Medicare funds, which includes almost all US hospitals, had to provide care to anyone presenting to its ED regardless of ability to pay [66]. The role of the ED in the US was officially expanded with this legislation to become a safety-net access point for healthcare for patients who have no health insurance or ability to pay for services. EMTALA is believed to be one reason why some patients access EDs rather than more appropriate lower levels of care.

More recently, changes in hospital same-day direct admit practices may have led to higher ED demand. Direct admissions occur when a primary care physician makes a request to the hospital to admit a patient. Morganti et al. [54] found that direct admissions to US hospitals for non-elective care decreased by 10% between 2003 and 2009. The physicians in their study cited increased barriers and time needed to coordinate a direct admit as reasons why they tell their patients to go to the ED instead of requesting a direct admit.

*ED Patient Flow.* Figure 2 shows the typical steps of ED patient flow. The first steps entail registration, triage, and waiting if a treatment room is not available. The underlying queueing system is a multi-server priority queue, with priority determined by the ESI level described in the next paragraph.

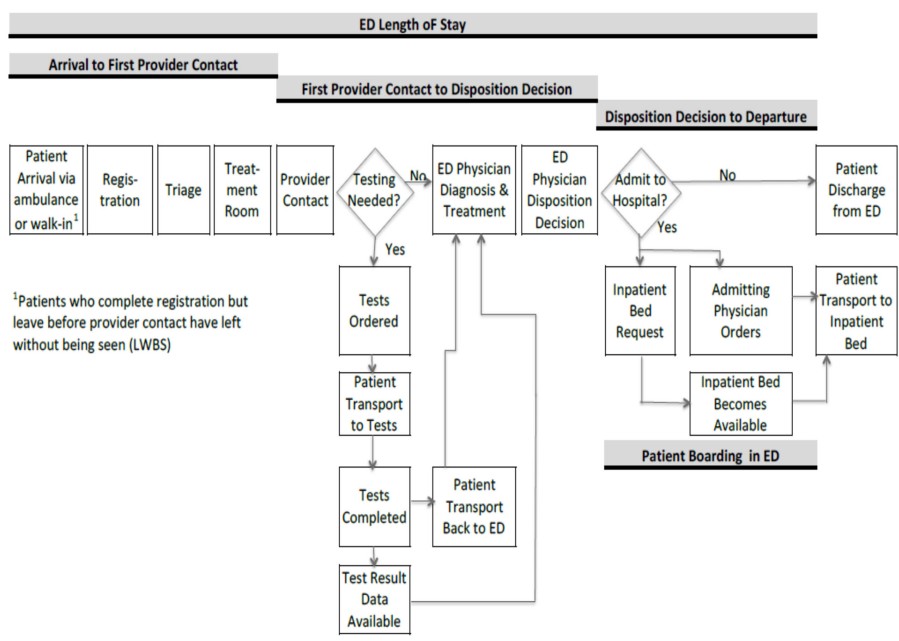

**Figure 2.** ED patient flow.

ED triage is the process in which a nurse assigns a priority rank to each patient. This rank determines the patient's service order and provides some information about the types of resources that the patient might need. In some cases, triage is used to route some low-urgency patients to a special area called a fast track. There are numerous priority scales in use internationally [67]. In the United States, a five-level triage tool called the Emergency Severity Index (ESI) is commonly used. The five levels are described in Table 3 [68].

Table 3. Emergency Severity Index (ESI) Levels.

| ESI Level | Description |
|---|---|
| ESI Level 1 | The patient requires immediate life-saving intervention (1–3% of all ED patients) |
| ESI Level 2 | The patient should not wait to be seen if they are in a high-risk situation, are confused, lethargic, or disoriented, or are in excruciating pain or distress (20–30% of all ED patients) |
| ESI Level 3 | The patient is not Level 1 or Level 2, has vital signs within the accepted range for the patient's age, and is predicted to require two or more resources, such as labs; diagnostic testing; intravenous fluids; intravenous, intramuscular or nebulized medications; specialty consultation; and/or a simple procedure or complex procedure (30–40% of all ED patients) |
| ESI Level 4 | The patient has vital signs within the accepted range for the patient's age and is predicted to use one resource. Levels 4 and 5 combined comprise 20–35% of all ED patients. Level 4 is an appropriate level to stream through the fast track. |
| ESI Level 5 | The patient has vital signs within the accepted range for the patient's age and is predicted to require no resources. Levels 4 and 5 combined comprise 20–35% of all ED patients. Level 5 is an appropriate level to stream through fast-track. |

Once a patient's turn arrives, he or she is cared for by a team consisting of an ED physician and a nurse. This team may also include consulting specialists, depending on the patient's needs. The ED physician diagnoses the patient's condition and orders tests as needed. Nurses often coordinate patient transfers to lab and imaging facilities. The ED exam/treatment room is held for the ED patient while he or she is away for tests. Hence, the exam/treatment room does not become available for a new patient until the current patient is either admitted to the hospital or discharged, which is called the disposition decision.

If the patient requires hospitalization, the patient's condition is usually stabilized first prior to transfer to the inpatient unit. ED physicians do not write admission orders because they typically do not have hospital admitting privileges [69]. Hence, patients requiring admission must also be seen by the admitting physician, generally a hospitalist, who writes the admission orders. Some patients may be placed on 24 h observation status before deciding whether admission is warranted. Some hospitals have separate observation units, whereas others place the patient in an inpatient unit bed. A patient may be boarded at any stage of the flow described in Figure 2 if resources necessary for the next step in the treatment are not available.

*Performance Metrics.* Key ED performance metrics relate to either baulking, flow time, or capacity. We list key metrics in each category below.

1.  Balking-related metrics: include patients leaving without being seen (LWBS) and ambulance diversions [63].
2.  Time-related metrics: time-to-treatment, time-to-treatment for specific medical conditions [64,65], ED length of stay (LOS) that exceeds some threshold, and ED boarding time [63].
3.  Capacity-related metrics: frequency of ED census approaching or exceeding the available ED beds or personnel capacity, the daily number of ED visits exceeding a targeted number, and ED nurses or physicians reporting being rushed [66].

ED performance issues have become so acute from both clinical and political standpoints that national mandatory reporting of key performance metrics has been implemented in some countries. The United Kingdom, for example, had, at one time, a national target of a maximum ED length of stay of 4 h. Such measures have since been dropped in favor of Clinical Quality Indicators (CQI) [45]. We describe these indicators in Table 4.

In the US, the Centers for Medicare and Medicaid Services (CMS) requires hospitals to report on five ED performance metrics [32,67]. As seen in Table 4, there are some similarities between the CMS' and UK's Quality-in-Emergency-care-Dashboard (QED) metrics. There are some notable differences as well. For example, CMS has a metric focusing on ED boarding, whereas QED tracks the timeliness of initial assessment for patients arriving by ambulance and repeat visits within 7 days.

**Table 4.** Summary of Key ED Performance Indicators.

| *QED* Indicators ([63], p. 31), (CQI; United Kingdom) | *CMS* Indicators McHugh et al. ([67], p. 5), (United States) |
| --- | --- |
| Time in the ED—% less than 4 h | The median patient time from ED arrival to ED departure for patients who were discharged |
| % of patients with ED stay exceeding 6 h | Median time from ED arrival to ED departure for admitted patients |
| Time for arrival to treatment by a decision-maker—% within 60 min or less | Door-to-diagnostic time, i.e., time to evaluation by a qualified medical professional |
| % Left Without Being Seen | The patient left before being seen |
| % Unplanned re-attendance to the ED within 7 days | No equivalent metric |
| No equivalent metric | The average amount of time admitted patients spend between being accepted and leaving |
| Time to initial assessment for patients arriving by ambulance—% less than 15 min | No equivalent metric |

ED performance is driven by daily cyclical variability in both ED arrivals and the availability of inpatient beds. Figure 3 shows the typical hourly variability in ED arrivals, for example, in hospitals. At this hospital, there is a sharp increase in arrivals around 8 AM. The average arrival rate remains high throughout the day and early evening. Also, Monday mornings have higher ED demand than other weekday mornings. Similar arrival patterns are observed in many other hospitals.

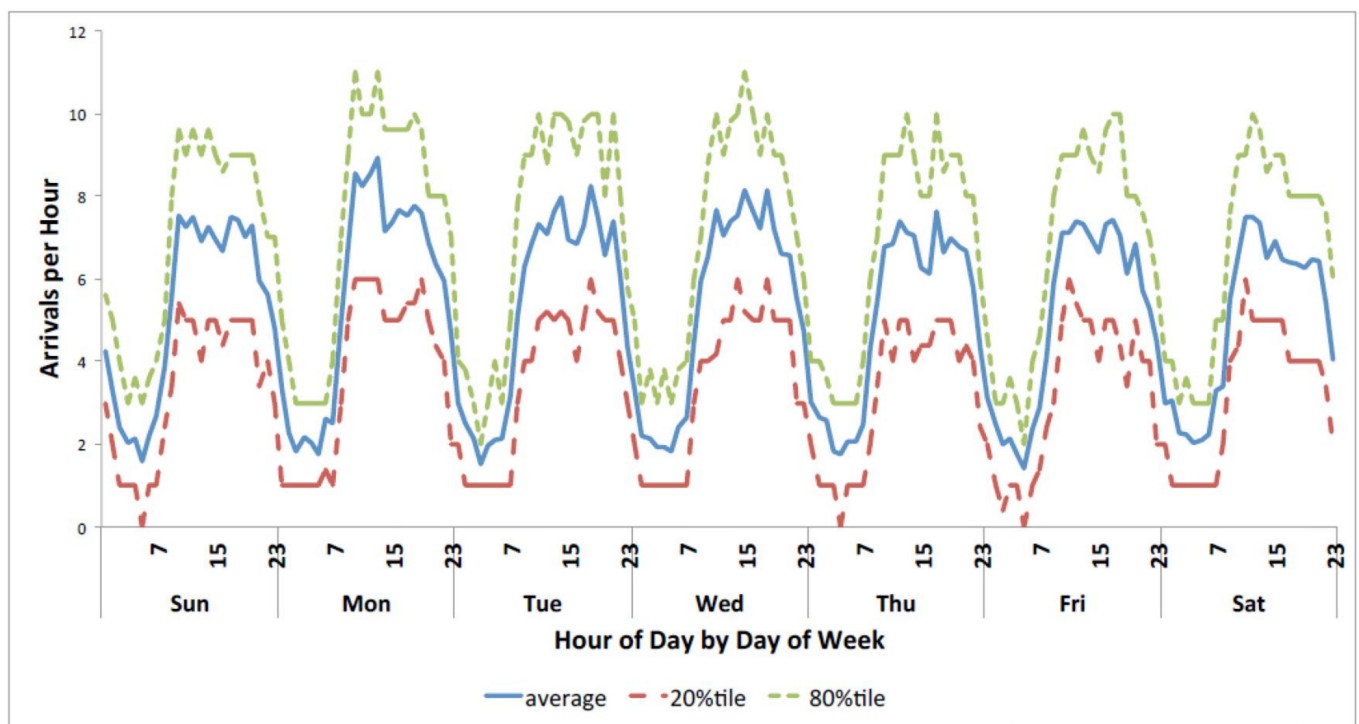

**Figure 3.** ED hourly arrival patterns.

Some of the ED patients require hospitalization. As described in previous parts, the weekday morning demand for inpatient beds by ED patients tends to build concurrently with the peak daily demand by post-surgery OR patients. This can result in high demand for beds that serve as the downstream location for both ED and OR patients. Figure 4 shows one month's data for one such hospital unit. Admission totals are the highest between 11 a.m. and 4 p.m., with great demand from both the ED and ORs. Later in the day, the bulk of the bed requests come from the ED.

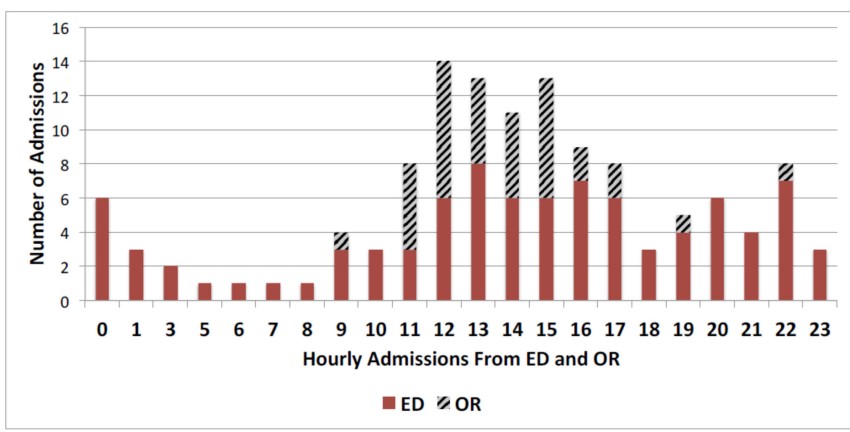

**Figure 4.** One month's total hourly inpatient bed requests from ED and OR.

Peaks in inpatient bed demand can result in patients boarding in the ED. We obtained an estimate of ED patient boarding time at one hospital. These data are summarized as the cumulative distribution function (CDF) "*This is related to probability Theory and statistics*" of ED board times and shown in Figure 5. The figure shows that 50% of the patients who experience boarding have to wait for more than 90 min, and 10% wait more than 3 h.

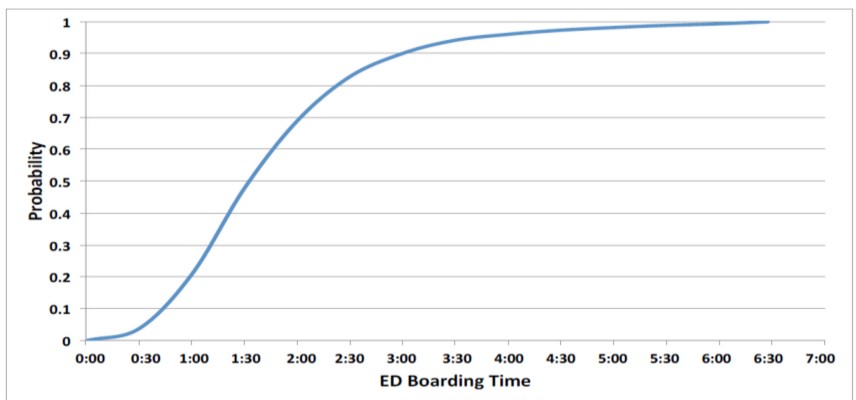

**Figure 5.** Cumulative distribution function of ED boarding times.

EDs cannot turn away walk-in patients because of EMTALA. Thus, ambulance diversion of emergency patients has been the primary strategy used by EDs to reduce arrivals during times of ED and inpatient bed congestion. However, if other EDs in the region are simultaneously near capacity, the increase in patient arrivals to the remaining hospitals from the diverted ambulances can worsen their congestion. This can result in additional EDs going on divert status, making it difficult for the ambulance personnel to find an ED with available capacity. Diversion also results in potential lost revenue for the hospital. How to prevent ambulance diversions and how to assign patients among available EDs when one or more EDs go on divert remain ongoing challenges. In addition, once a hospital is on divert status, it is not clear what policies should guide its decision to go off the divert status.

### 3.2. ED Treatment Process

Operational challenges in the ED treatment process include long wait times, excessive lengths of stay, patient boarding, and overcrowding. EDs must cope with patients that require significantly different services (diagnostic, physician, and inpatient care) and have different urgency levels. The role of the ED is to stabilize patients' conditions so that they may be transferred to other appropriate parts of the healthcare system (inpatient care or follow-up primary care) or discharged home.

Performance metrics focusing on efficiency and throughput mirror this view. But the role of the ED is evolving. Advanced diagnostic testing and aggressive treatment in EDs can prevent the need for hospitalization. In addition, diagnostic workups and access to specialists that transpire over days or weeks in the traditional outpatient setting can be completed within hours in the ED [68,69].

In this role, the ED also serves as another delivery node in an ongoing episode of care. This requires more effective ED patient discharge planning and better care coordination with other providers across the healthcare service chain (Johns Hopkins University, Armstrong Institute for Patient Safety and Quality, 2014 (https://www.hopkinsmedicine.org/armstrong_institute/, accessed on 20 May 2021)). Care coordination is especially important for chronic disease care, which accounts for about 75% of US healthcare expenditures [70]. The quality and effectiveness of ED care within that episode must be quantified. Thus, the ED operational regime is changing from one that has been primarily efficiency-driven to one that increasingly requires balancing quality and efficiency. EDs are developing new triage and patient streaming methods in response to their changing roles.

### 3.3. Downstream Bed Availability

As discussed before, many countries have reduced their supply of acute care beds, resulting in higher bed occupancy rates. The lack of inpatient bed availability has been implicated as a driver of poor ED performance [43,71,72]. In the US, the lack of intensive care unit (ICU) beds specifically is frequently cited as a primary reason for ED boarding [73].

Higher bed occupancy makes it more difficult to respond to variability in patient census and ADT. Physician daily rounding practices affect the timing of nursing work activities related to bed management. Some physicians round first on the most critically ill patients in order to prioritize entering their test and treatment orders. This can delay discharge orders for patients who could be discharged that day, exacerbating daily bed shortage situations. If facilitating patient discharges is a priority, then rounding first on patients most likely to be discharged allows nurses to focus first on patient discharges. However, rounding first on patients most likely to be discharged delays that day's ongoing diagnosis and treatment for remaining patients, which could increase their overall hospital lengths of stay.

A key point of the discussion so far is that ED performance improvement requires multi-faceted intervention inside and outside the ED. Upstream, it requires shaping patient behavior and improving primary care capacity to try to reduce unnecessary ED utilization. Downstream, it requires improving patient flow throughout the entire hospital, not just the ED. And given OR, ED, and inpatient demand patterns, the bottleneck resource affecting efficient ED flow may fluctuate over time. Finally, as the role of the ED evolves to provide timely access to advanced diagnostic technology and specialist care, a new patient type is emerging—one who needs immediate care, who will require extensive diagnostic or treatment time in the ED, and whose ED visit can likely prevent the need for hospitalization. We turn to discuss how EDs are coping with these operational challenges.

## 4. OM Opportunities

OM opportunities for ED demand–supply matching fall into two broad categories: (1) those that focus on the hospital walls and (2) those that focus on new outpatient care models that are expected to reduce ED and inpatient demand. In this and the next paragraph, we discuss the first category. Within the hospital, OM models have identified useful strategies that could be translated into computerized applications that support real-time decision-making by providers. For example, the statistical work of Peck et al. [74,75] on predicting hospital admission could be combined with the streaming models of Saghafian et al. [76] to support virtual streaming in real-time. Similarly, the ED wait time prediction models for individual hospitals in an EMS region could be integrated into ambulance routing policy models to alleviate ambulance diversion issues.

OM models, combined with real-time data feeds, have the potential to support providers' real-time decision-making, resulting in superior ED workload management strategies. The data sources for these models reside in the timestamps of computerized scheduling, transport, and medical record systems. New, richer timestamp data sources are coming online as some hospitals implement real-time radio frequency identifier tags on patients, staff, and equipment that transmit location information every few seconds to a database [62,77–80]. Plambeck et al. [81] highlight the operational challenges of using real-time data feeds to support OM models.

OM models that address new approaches to design outpatient chronic disease care will need to focus on the entire healthcare service chain. Some OM models that have this characteristic focus on only one chronic disease condition, but patients typically have two or more chronic conditions. OM models for chronic disease management will need to take this into account. The data sets needed to study these models are much more labor-intensive to build, as they span multiple locations, e.g., primary care clinics, EDs, urgent care clinics, hospitals, nursing homes, home health agencies, etc. It is difficult to get complete patient data across episodes of care because the patient may use services that are not owned by a single health system, and insurance companies will only have data on those services that they cover. Building and analyzing these types of data sets will require closer collaboration between OM modelers and health services researchers with expertise in merging large patient data sets.

The most readily available data from hospital databases concerns bed occupancy (or census) levels in each unit. With bed management software, it is possible to track every activity: admission, transfer in, transfer out, and discharge. These data can be used to understand how the census changed over time. In Figure 6, we show the combined census from four interchangeable units of a medium-sized hospital. We examined the total census of approximately interchangeable units to understand demand variability at the level of patient type, where patient type is defined by their nursing needs. Figure 6 shows significant variability, which led us to look for possible explanations for this phenomenon.

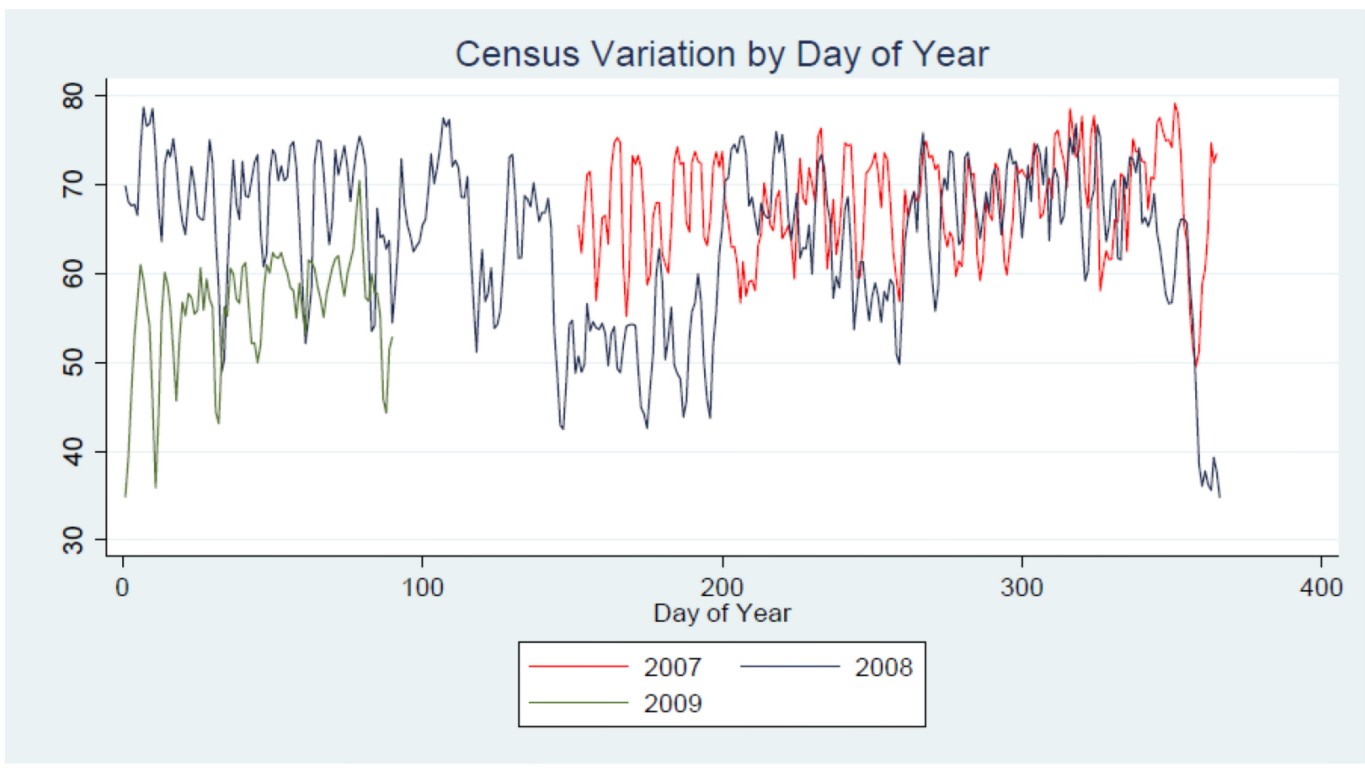

**Figure 6.** Census fluctuations across four partially interchangeable units.

In Figure 7, we show how the start-of-shift census in a single unit changes by day of the week. There is a clear buildup of census until Thursday and then a sharp decline on Saturday. In Figure 8, we show admissions, transfers in, transfers out, and discharges, which help explain these changes in the census. For example, there are more transfers from OR (shown in dark solid line) on Mondays, Tuesdays, and Wednesdays, whereas discharges (shown in dark dashed line) peak on Fridays. In contrast, transfers from other units remain consistent across weekdays. The direct admit is smaller on Saturdays and Sundays. These patterns are driven by the typical workweek cycle.

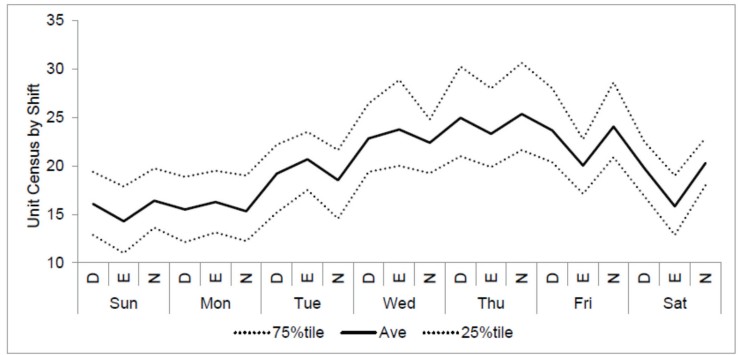

**Figure 7.** Start-of-shift census fluctuations in a single unit—weekly cycle. D = day shift, E = evening shift, and N = night shift.

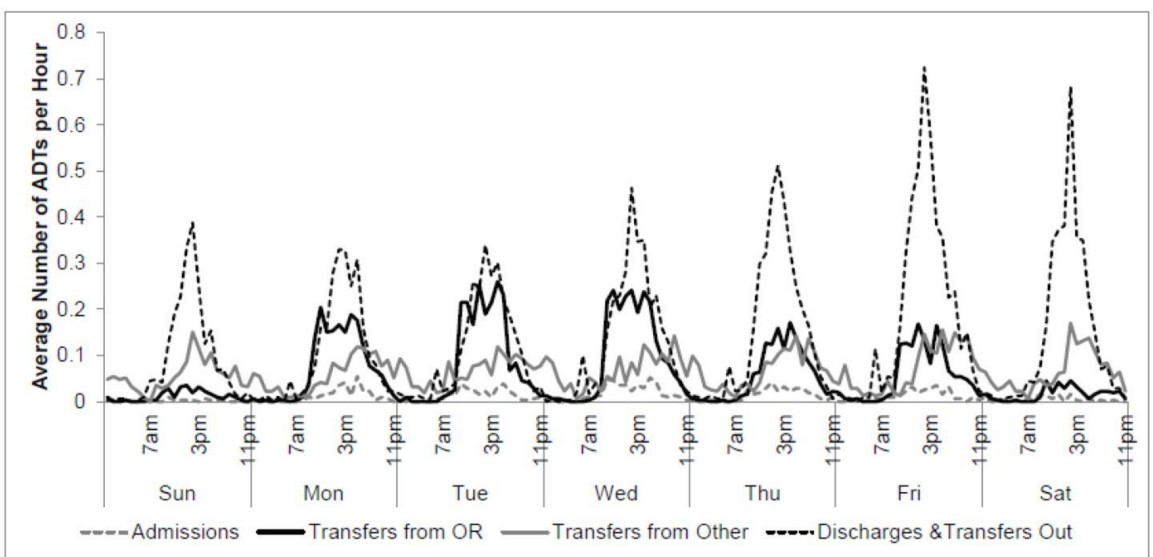

**Figure 8.** Weekly activity: admissions, transfers in, transfers out, and discharges.

In Figure 7, we observe that census tapers off toward the end of the week, suggesting a relatively lower demand for staffing. However, when we examined nurse-to-patient ratios (the nurse-to-patient ratio refers to the number of patients in the unit per nurse and is usually calculated based on start-of-shift census and nurse availability) across different shifts and compared them to the planned ratios (the planned ratio for a unit refers to the shift-specific nurse-to-patient ratio that is acceptable both to the management and to the nurses' union), we find the largest discrepancies during the night and weekend shifts; see Figure 9. The planned nurse-to-patient ratios for weekend and night shifts are higher, which means fewer nurses are scheduled to be on duty. In addition, the census usually drops during weekends, which further lowers the number of nurses that are scheduled to work on weekends. This means that weekend availability is at a greater risk of being compromised even if a small number of nurses are absent. Figure 9 shows the distributions of realized nurse-to-patient ratios after accounting for planned and unplanned absences. The S-shaped

curve is the cumulative distribution function of the nurse-to-patient ratio, the black dotted line is the average observed nurse-to-patient ratio, and the red line is the planned ratio. Note that even for those shifts in which the planned and average observed nurse-to-patient ratios are quite close, the distribution of these ratios shows significant variation. That is, there are frequent cases of both under and overstaffing. Hospital managers often concentrate on the average of realized nurse-to-patient ratios. What the above analysis shows is that nurses are subjected to significant variability in their workloads even when the average workload equals the planned levels. Specifically, for the hospital whose data were used to develop Figure 9, there is a frequent shortage of nurses on the night shift—both during weekdays and weekends. (The demand for night shift nurses will never decrease because healthcare is available around-the-clock, and many nurses consider the evening shift to be a desirable and reliable option. The Bureau of Labor Statistics reports that 4% of Americans who worked in 2017–2018 did so on a night shift. In order to increase the safety of patients and healthcare workers, organizations must assess practices and policies to reduce the inevitable fatigue that results from long night shifts. Alternative shift lengths may be investigated, or authorized workplace naps may be an option).

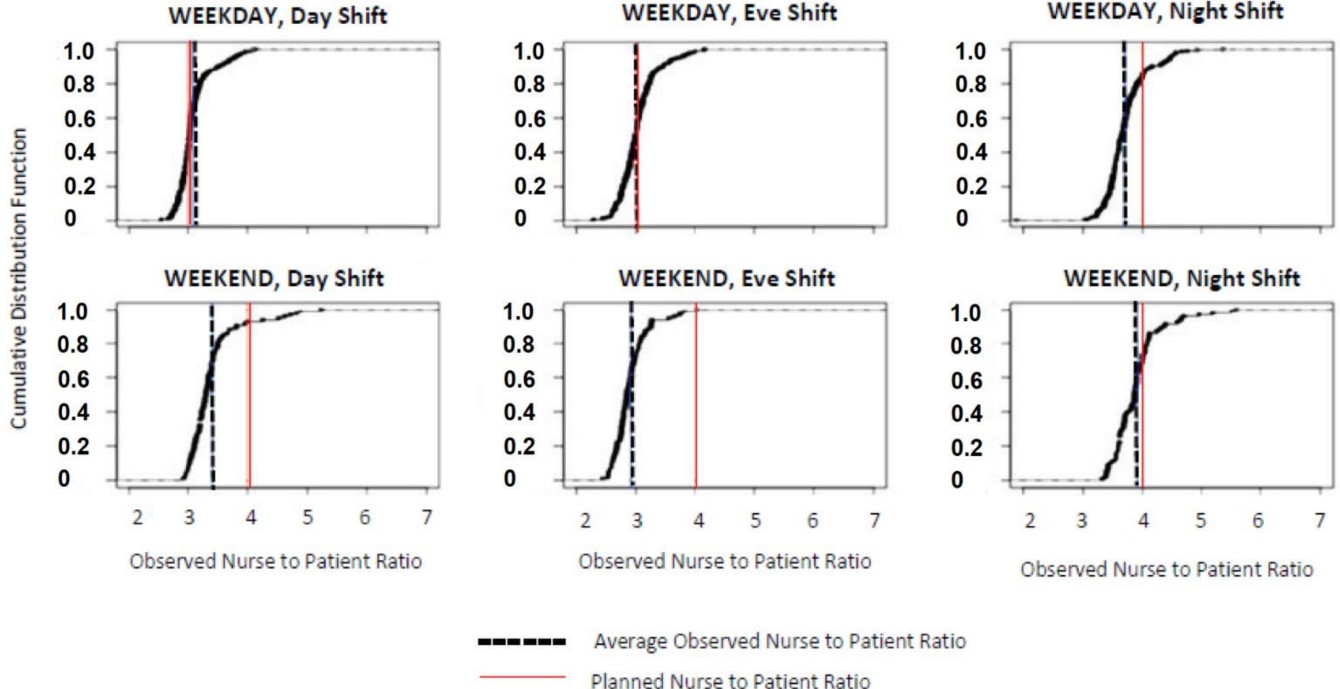

**Figure 9.** Nurse-to-patient ratios, actual and norms by shift and by weekday.

Because nurse staffing levels affect the welfare of both patients and nursing staff, federal and state legislators have placed certain requirements on hospital administrators' decision processes. For example, according to the American Nurses Association (2015), Federal Regulation 42CFR 482.23(b) requires "*hospitals certified to participate in Medicare to have adequate numbers of licensed registered nurses, licensed practical (vocational) nurses, and other personnel to provide nursing care to all patients as needed* [82]." Several states have enacted legislation to require hospitals to have either one or more of the following measures: (1) nurse-driven staffing committees, (2) legislated and/or regulated minimum nurse-to-patient ratios, and (3) disclosure of staffing levels. Specifically, fourteen states (CA, CT, IL, MA, MN, NV, NJ, NY, OH, OR, RI, TX, VT, and WA) currently mandate minimum nurse-to-patient ratios in either law or regulation.

Lang et al. (2004) presented a systematic review of 43 papers on the effect of mandated minimum nurse-to-patient ratios in acute-care hospitals [83]. In this review, patient outcomes were measured only in terms of in-hospital adverse events. The authors concluded

that fewer patients per nurse were linked to better outcomes, specifically, shorter hospital stays, lower inpatient mortality rates, and lower failure-to-rescue rates. However, there was no evidence in favor of specific minimum nurse-to-patient ratios. OM studies that focus on the quality-of-care and staffing cost tradeoffs could significantly add to the debate on what remains an open question: should either state or federal governments require minimum nurse-to-patient ratios, and if so, what levels should be prescribed?

Evidence from data reveals significant challenges in managing staffed bed capacity due to variability in both supply and demand. The problem is further exacerbated by specific skill sets required to provide care to specific patient types. Among the possible sources of variability, it is possible in some cases to identify some as natural and others as artificial. For example, variability in the number of admissions to a unit is largely natural, whereas variability in transfers from OR is driven by OR schedules. However, for the most part, it is difficult to tease out whether variability is a consequence of naturally occurring changes in demand and supply or those caused by specific actions of hospital managers. Still, when possible, it will make sense to exploit knowledge of OR schedules to better plan staffed-bed capacity and to jointly plan OR schedules and staffed-bed capacity.

### 5. Impact of Unit Size & Scope

The first example concerns the size of nursing units in terms of bed capacity for a particular care intensity. We assume that the hospital has determined the total number of beds needed to serve its patient population, and what remains to determine is the number of interchangeable units and the size of each unit. These units will be staffed with the same skill-level nurse mix, equipped with the same in-room equipment, and staffed based on the same average nurse-to-patient ratios. Also, none of these units is a swing unit.

Let $m_j$ be the number of nurses that can be supervised by a single charge nurse (the lead nurse in each shift is referred to as the charge nurse and makes nurse-to-patient assignments and oversees admissions, discharges, and transfers) in shift $j$. In addition, let $r_j$ and $(r_j - \hat{r}_j)$ be the average number of patients that can be cared for by a regular nurse and a charge nurse, respectively, in shift $j$. Parameters $r_j$s and $(r_j - \hat{r}_j)$s are called nurse-to-patient ratios. Typically, morning and evening shifts have the same ratios, but the ratios are different for the night shifts. Then, the bed capacity that allows maximum nurse productivity to be achieved in all shifts is a multiple of $m_j r_j + (r_j - \hat{r}_j)$ for each $j$. For example, if $m_j = 8$, $r_j = 3$ *and* $r_j - \hat{r}_j = 0$ for each $j$, then unit size in multiples of 24 allows maximum productivity of nurses. If this hospital were to build a unit with 22 beds (say), then the nurses' productivity may be lower because of greater indirect costs and because when the unit is full, the eighth nurse will serve only one patient rather than three.

In the above example, it was straightforward to determine an efficient bed capacity because the nurse-to-patient ratio did not vary by shift. For a different example with $m_1 = m_2 = m_3 = 7$, $(r_1, r_2, r_3) = (3, 3, 4)$ and $(\hat{r}_1, \hat{r}_2, \hat{r}_3) = (3, 3, 1)$, the most efficient bed capacity is 651. This is not a realistic size for a nursing unit in most hospitals.

In fact, in a hospital that the authors are familiar with, there are four interchangeable units, three with 22 beds and the fourth with 25 beds, with the nurse-to-patient ratios mentioned above. The bed capacities of these units do not allow maximum productivity to be achieved in either day, evening, or night shifts.

Formally, for a unit with bed capacity $i$, the number of nurses required when $x_i \leq k_i$ beds are occupied is $\left[(x_i - n_0(r_j - \hat{r}_j))/r_j\right] + n_0$, where $n_0$ is the smallest number of charge nurses needed to supervise other nurses in the unit ($i.e.$ $n_0 = \{\min n : nm_j \geq \left[(x_i - n(r_j - \hat{r}_j))/r_j\right]\}$). Having these arguments in hand, consider two structures: (a) $u$ units each with bed capacity $k_i$, and (b) a single large unit with bed capacity $k = \sum_{i=1}^{u} k_j$. It is clear that with each fixed level of bed occupancy $x$, an optimal patient placement policy must assign patients in multiples of $m_j r_j + (r_j - \hat{r}_j)$ to the extent that is possible. Moreover, since each unit with at least one assigned patient must have a charge nurse, the number of charge nurses in an arrangement (a) is no less than the number

of charge nurses in the arrangement (b). Put differently, if $x \leq k$ patients are assigned optimally, structure (b) allows at least as good or better maximum productivity.

The above arguments lead to the conclusion that a single large unit dominates combinations of smaller units of the same bed capacity. The extent to which many smaller units could lower productivity depends on the nurse-to-patient ratios and the number of such units. This result is based on the assumptions that (1) nurse-to-patient ratios are fixed (in reality, acuity is different for each patient and changes over time as the patient's medical status changes), and (2) there are no physical constraints (such as the location of nursing stations and limited floor space) on the design of nursing units. When these assumptions are relaxed, the problem of determining dominant structures is more complex. The determination of dominant design is further complicated by the presence of permanently open units and one or more swing units. The latter are closed/opened as needed based on patient volumes. In this environment, the determination of the bed capacity of permanently open and swing units is an open problem.

Some hospitals have come up with strategies to provide additional flexibility by creating a small unit-within-a-unit. In an example of such arrangements, a step-down unit may have a few beds placed in special rooms, which can be *stepped up* to act as ICU beds when needed. A few nurses in the unit are specially trained for such purposes, and the rooms are equipped with more monitoring devices. The beds are staffed at the nurse-to-patient ratio that is normal for the unit for regular patients and typically at a 1:1 ratio for ICU patients. The step-up beds are used to create room in ICUs by transferring patients who are relatively well but still in need of intensive care. The authors are familiar with hospitals that have successfully implemented this approach for managing their ICU bed needs.

Finally, the effectiveness of size and scope decisions is affected by choice of performance metrics. In our discussion so far, we considered only the maximum achievable productivity. Other measures may be appropriate, e.g., average overtime costs, patient waiting costs, and less-than-ideal placements or turn-away (e.g., via ambulance diversions).

*5.1. The Impact of the Choice of Performance Metrics*

Hospitals typically evaluate a unit's performance by calculating variance, i.e., the difference between the budgeted and realized NHPPD (developed by the ANA for the NDNQI, NHPPD refers to the number of nursing care hours provided divided by the number of patients in that hospital unit during a 24 h period) in each review period or revenue per nurse FTE (FTE stands for full-time equivalent) [68]. Note that the meaning of variance in nursing literature is different from its meaning in probability and statistics. The length of a review period is typically a month, but it could be as small as a shift or as large as a quarter [68]. The budgeted amounts are based on historical averages, NHPPD comparisons with other similar hospitals, and realized patient census. Variance analysis is used to control expenses, and unit managers are held accountable for using resources in excess of the budgeted amounts. However, as we show next, a variance-based review of unit managers' performance may lead to poor operational choices.

Consider a short-stay nursing unit that is staffed to provide k nursing hours of care per day. Given k available hours, suppose *P(k)* is the max number of patient days of care that the unit can generate per day. We assume that *P(k)* is increasing in k. However, because the output is constrained by daily demand *D* (expressed in patient days), the actual number of patient days of care produced is *min{P(k), D}*. Let $F(\cdot)$ and $\overline{F}(\cdot)$ denote the *CDF* and the *CCDF* (complementary cumulative distribution function) of *D*, and b denote the benchmark number of patient days per nursing hour used to calculate the unit's productivity. Then, a nursing unit manager who is evaluated against this benchmark tries to minimize $E[(bk - min\{P(k), D\})^+]$. Clearly, $k = 0$ is a minimum, but that is not a practical solution. So, as an alternative, a unit manager may choose a target staffing level based on tolerance $\delta$ for maximum deviation from the benchmark. That is, a possible target staffing level is

$$\overline{k} = max\left\{ k : E[(bk - min\{P(k), D\})^+] \leq \delta \right\}.$$

Next, we consider a different performance criterion for the problem. The hospital incurs a cost $c$ per unit of nursing care provided, a penalty $p$ per turned-away patient day, and a revenue $r$ per patient day served. Therefore, operational choice of staffing level should depend on $\pi(k)$, where $\pi(k) = -ck - pE[(D - P(k))^+] + r E[\min\{D, P(k)\}]$. When $P(k)$ is concave, it can be shown that $\pi(k)$ is concave in $k$ (proof is omitted in the interest of brevity), and an optimal staffing level $k^*$ satisfies $p(k^*) = \overline{F}^{-1}(c / [(r + p)p'(k^*)])$. Clearly, the $k\_$ calculated above is not in general equal to $\overline{k}$ calculated earlier, leading to systematic choices of uneconomical staffing levels.

Performance metrics such as NHPPD view nursing care as a source of cost and penalize nurse managers who do not keep costs within mandated ranges. The point of view that nurses are a source of cost touches a nerve with nurses' unions, who argue that nurses also produce healthcare [70]. Therefore, both costs and outcomes need to be considered at the same time. Numerous articles in the literature on nursing care make the case that a shortage of nurses endangers the standard of care and patients' safety, lengthens hospital stays, and decreases nurses' job satisfaction (e.g., [31,83–90]).

*5.2. The Impact of Patient Movement Policies*

In this section, we present two simple concepts. First, we discussed that if a hospital were to choose between floating nurses or moving patients, and neither had any negative consequences on patients' health outcomes and neither consumed nurses' time, then moving patients would always be superior to floating nurses. This comes from the fact that a nurse typically takes care of more than one patient. The second concept we present is that if a hospital has a consistent patient movement policy and it chooses nurse staffing levels commensurately, then the choice of a particular policy does not significantly affect performance metrics of interest. The latter is based on experiments performed via a computer simulation. We chose policies that are consistent with practices one would find at many hospitals. The details follow.

Suppose a hospital has two units that belong to the same care intensity hierarchy, and nurses in each unit are trained to take care of patients with similar diagnoses and care needs. Suppose $y_i$ nurses are assigned to unit $i$ in each shift, and the nurse-to-patient ratio is 1:$k$ in all shifts. If, at any time, the number n of patients needing beds is less than $k(y_1 + y_2)$, then this demand can be accommodated without requiring additional nurses. Depending on the starting census levels in each unit, this may involve either patient movement or float of nurses. That is, when nursing needs can be met by the available cohort of nurses, it should make no difference to the hospital whether it moves patients or floats nurses, under the assumption that neither induces additional cost to the hospital.

Given inpatient units within the same care-intensity hierarchy, a hospital may place patients in any one of these units and provide equal care quality. The hospital could choose from a plethora of patient movement policies. For example, it could have a hierarchy of movement priority: patients move to the highest priority unit first, then to the next higher priority unit, and so on. We call this 'static arrangement priority' because the highest priority unit fills up first, then the next higher priority unit, and so on. Compare this to a different policy, which we call targeting, in which patients are placed to keep census constant across similar units. We then tested for which of these two policies is likely to perform better. An example of the two approaches is shown in Figure 10. In this example, there are two equal care-intensity and equal bed capacity units, and we illustrate the placement of five new patients on a given day.

We developed computer simulation models of each placement strategy under two scenarios. In the first scenario, nurses could be floated from one unit to another, whereas in the other, nurse floating was not permitted. In each case, the number of nurses assigned to each unit was chosen to match the anticipated workload. For example, in the static priority strategy, the highest priority unit was staffed with enough nurses to accommodate a full census, whereas, in the targeting strategy, each unit received an equal number of nurses.

We provide complete details of the parameters used to set up the simulation model in the ensuing paragraph.

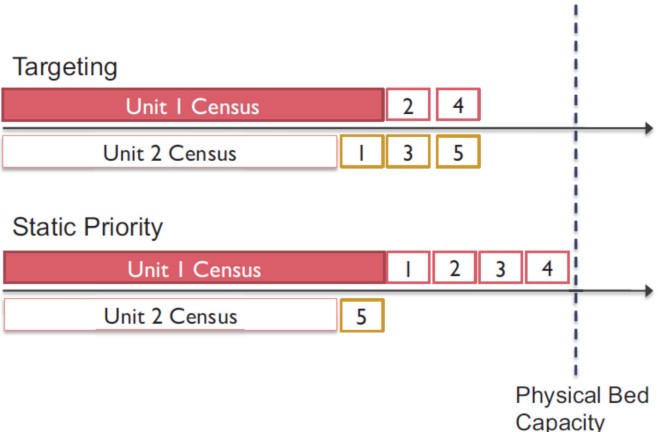

**Figure 10.** Two possible placement strategies.

In the simulated experiments, we assume that the hospital has complete information about whether a patient will leave the unit within an hour or not. The planned staffing level is at a level that is sufficient for the average census in the unit. Nurses can float to other interchangeable units only at the beginning of a shift. Overtime (OT) is calculated after utilizing all available float nurses. Specifically,

$$\text{minimum OT requirement} = \text{nursing needs of patients carried over from the previous shift} - \text{staffing level} \tag{42}$$

The patient is not to be moved from one unit to another once admitted to a particular unit. Patients may be placed into a unit so long as there is still bed capacity. When staffed-bed capacity is reached, the patients in excess of the staffed-bed capacity are counted as unserved patients. Fractional OT shifts are allowed.

The parameters used in our discrete-time simulation are as follows: 3 units, 30 beds each; nurse-to-patient ratio = 1:3; hourly patient arrival rate = 1.17 (Poisson distributed); average LOS = 61 h; LOS distribution. Example 1: Poisson (61), Example 2: Geometric (1/61); nurse absenteeism probability p = {0, 0.1, 0.2}. Note that the two lengths of stay distributions have the same mean but different variability. The aggregate staffing level was 80% of bed capacity, i.e., $(0.8 \times 90)/3 = 24$ RNs. The arrival rate was chosen such that the average number of patients in the system was approximately 72. Under targeting strategies, each unit is staffed with an equal number of nurses. Under a static priority strategy, first, the two highest priority units are staffed fully with 10 nurses each, and then the remaining 4 nurses are assigned to the lowest priority unit. Thus, staff assignment strategies match patient movement strategies.

Performance is measured by calculating the minimum OT requirements, the number of unserved patients, and the frequency with which nurses may need to be floated from one unit to another. The average performance metrics obtained from these experiments are shown in Table 5. Note that the average statistics on minimum OT per shift, the number of unserved patients per shift, and the number of nurses that need to be floated per shift (movement) are statistically not different across these strategies at a 95% confidence level. This suggests that so long as the patient placement strategy roughly matches the staff assignment strategy, the choice of movement strategies does not have a significant effect on the performance. This result is satisfying from a practitioner's viewpoint because it allows hospital managers to choose from a variety of different options depending on their individual circumstances.

**Table 5.** The relative performance of targeting versus static priority strategies.

| Absentee Rate | Metric | Poisson LOS | | Geometric LOS | |
|---|---|---|---|---|---|
| | | Targeting | Static Priority | Targeting | Static Priority |
| 0% | Min OT/Shift | 0.83 | 0.95 | 0.54 | 0.61 |
| | # Unserved/Shift | 0.74 | 0.65 | 0.49 | 0.43 |
| | Movement | 0.43 | 0.4 | 0.34 | 0.25 |
| 10% | Min OT/Shift | 2.37 | 2.38 | 1.75 | 1.74 |
| | # Unserved/Shift | 1.17 | 1.21 | 0.86 | 0.92 |
| | Movement | 0.75 | 0.49 | 0.87 | 0.57 |
| 20% | Min OT/Shift | 4.22 | 4.18 | 3.53 | 3.48 |
| | # Unserved/Shift | 1.45 | 1.58 | 1.21 | 1.34 |
| | Movement | 0.61 | 0.39 | 0.76 | 0.45 |

In the next section, we discuss the state of OM research dealing with inpatient bed capacity. Key inputs to such models are estimates of distributions of demand for inpatient beds and lengths of stay (LOS). There are many papers in the health services literature that analyze data to estimate these critical parameters. Papers in the OM literature tend to assume that such estimates exist.

## 6. Conclusions

Spending on inpatient care across OECD countries accounts for almost 30% of overall healthcare expenditures, with outpatient care consuming another 33% [91]. Although spending growth rates have slowed in recent years, spending is still increasing. It is anticipated that the cost pressures of GDP spending on healthcare will continue to grow as the world's population ages. Spending increases are counterbalanced by a global life expectancy increase of about six years in the last 20 years, which has been attributed to advances in healthcare (GBD "*The Global Burden of Disease Study*", [92,93]).

Efforts to reduce healthcare costs have focused on cutting hospital beds, cutting salaries of healthcare workers, reducing reimbursement to healthcare providers, cutting the healthcare workforce, and increasing patient responsibility for covering their personal healthcare costs through co-payments [94,95]. However, trying to shape supply and demand for healthcare services using these approaches has its limits, and their unintended consequences can be counterproductive. The recent Medicare hospital payment penalties for 30-day hospital readmission rates (see [96]), for example, were implemented to incentivize improved hospital discharge planning and better patient handoffs from inpatient to outpatient care. What has happened instead is that many hospitals are readmitting patients for observation stays [97]. This reduces the hospital's readmission rate since observation status is not technically a hospital admission. However, observation stays can result in higher out-of-pocket costs for patients, especially if they require nursing home care after hospitalization. This is because Medicare does not cover nursing home care unless a patient has been admitted to the hospital for at least 3 days.

Healthcare organizations face many challenges as they adjust to reimbursement changes that are evolving from fee-for-service to value-related payments. Current and future reimbursement policies will continue to encourage healthcare organizations to focus on managing the care across the entire supply chain to reduce the need for escalation of care to the inpatient setting. These challenges highlight the need for OM models that help improve the efficiency and effectiveness of care for populations of patients across the entire supply chain of care, focusing simultaneously on throughput and outcomes and provider and patient perspectives. OM efforts to improve chronic care management highlighted earlier provide examples of ongoing work needed in designing and evaluating new care models [98–109].

Regardless of how care models evolve, however, there will always be a need for inpatient care. This monograph highlighted recent OM work that models the dynamic

inter-related impact of demand–supply matching across the ED, OR, and inpatient units. Healthcare managers and clinicians tend to make staffing and scheduling decisions across these areas independently of each other; as demand unfolds in real-time, clinicians and managers reactively firefight as best they can to reallocate staffing to where it is most needed at a given point in time to alleviate bottlenecks in patient flow. As OM researchers, we need to develop models that enable healthcare managers to simultaneously improve ED demand forecasting, OR scheduling policies, and medium- and short-term staffing plans that take into account the inter-relationship of how demand unfolds over time across the ED, OR, and inpatient units [100–109].

Where OM work has fallen short of its full potential is in implementing and testing models for real-time decision support tools. In the medical field, major research funding agencies have created a focus on translational research to more quickly bring new medical knowledge from the bench to the bedside. Some funds are targeted to specifically support such translational research. We recommend a similar focus on the OM fields. Newly developed OM models have the potential to aid healthcare managers in better matching demand and supply. The increasing use of electronic medical records, staffing systems, and scheduling systems contain relevant data to feed these models to support real-time decision-making. To date, we have not systematically focused on the challenges of bringing OM models from bench to bedside and then testing the impact of the models on improving the efficiency and effectiveness of care delivery. To be more relevant, we need to understand and appreciate how to embed OM models for decision support into the computer systems used by clinicians and managers and work collaboratively with them to achieve the full potential that OM has to offer.

Finally, we suggest four key nursing unit operational choices that influence supply–demand matching. These include decisions about unit size and scope, performance metrics, shift assignments and time-off requests, or what we refer to as HR policies (human resources policies), as well as decisions about patient movement. We will demonstrate that all four decisions have an effect on nurses' productivity and hospital costs, even though the unit size and scope choices are the most obvious as being pertinent to OM topics. Because (1) an estimated 80% of direct care costs in hospitals are payroll-related [31] and (2) nursing care has a dominant influence on care, we can justify our focus on nursing costs while ignoring other care providers.

**Author Contributions:** M.H., Formal analysis, Methodology, Software, Data curation, Writing—original draft preparation, Writing—review and editing, validation. K.K.L., Project administration; funding acquisition, investigation, supervision. Y.F., Visualization. X.L. Resources. All authors have read and agreed to the published version of the manuscript.

**Funding:** This research received no external funding.

**Institutional Review Board Statement:** Not Applicable.

**Informed Consent Statement:** Not Applicable.

**Data Availability Statement:** Not Applicable.

**Conflicts of Interest:** The authors declare no conflict of interest.

## Nomenclatures

A list of abbreviations used in this research is as follows:

| | |
|---|---|
| A&E | Accident and Emergency |
| ABM | Total Surgery Minutes |
| ADT | Admission, discharge, and transfer system |
| ANA | American Nurses Association |
| BES | Block Efficiency Score |
| CCDF | Complementary Cumulative Distribution Function |
| CCU | Critical Care Unit |

| CDF | Cumulative Distribution Function |
| CMS | Centres for Medicare and Medicaid Services |
| CQI | Clinical Quality Indicators |
| EDs | Emergency Department |
| EMTALA | Emergency Medical Treatment & Labor Act |
| ESI | Emergency Severity Index |
| FTE | Full-time equivalent |
| GBD | The Global Burden of Disease Study |
| ICU | Incentive Care Unit |
| LOS | Length of Stay |
| NDNQI | National Database of Nursing Quality Indicators |
| NHPPD | Nursing Hours per Patient Day |
| OM | Operations Management |
| ORs | Operating Room |
| OT | Overtime |
| QED | Quality-in-Emergency-care-Dashboard |
| SDS | Supply Demand Score |
| TDABC | Time-driven activity-based costing |
| TBM | Total in _ Block Minutes |
| TSM | Total Surgery Minutes |

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
