# Peer review of "A Review of Emergency and Disaster Management in the Process of Healthcare Operation Management for Improving Hospital Surgical Intake Capacity"

_mathematics, doi:10.3390/math10152784_

Round 1
Reviewer 1 Report
Page 1, Abstract: My suggestion is to include the most important findings of your research in the abstract.
Page 1, Abstract, Keywords: My suggestion is to include OR in the Keywords.
Section 1. INTRODUCTION: My suggestion is to include a subsection which will be dedicated to terminology (global definitions of basic concepts used in your research, e.g., OR, OM etc.) for the benefit of the readers who are not familiar with these concepts.
Section 1. INTRODUCTION: My suggestion is to include a Data Flow Chart describing all your methodological steps, so that the reader can obtain a clear overview of your work from the early beginning of the paper.
Section 1. INTRODUCTION, page 2, line 44, “…Often, ad-hoc approaches are used…”. My suggestion is to add some more examples together with their associated references.
Page 4, 2.1 NUMBER OF ORs AND FLEXIBILITY, lines 167-168, “…For making long-term capacity decisions, it is often useful to think of a week as a time unit…”. My suggestion is to include, within your manuscript, a full justification why one week is considered as an appropriate time interval to make “long-term capacity decisions”.
Page 7, 2.2 STAFFING DECISIONS, lines 305-306, “…We consider each historical day of data for a particular weekday (e.g., Mondays)…”: Did you consider all seven days of the week as weekdays? Is there a special reason to choose “Monday” ?
Page 5, line 216, “…It is customary to assume that patients’ wait cost and overtime costs are linear…”: Could you please provide, within your manuscript, some examples together with their associated references which will support your assumption that the specific costs are linear?
Page 13, 2.5 IMPROVING SCHEDULES, line 596, “…These decreases staffing costs…”: Is it possible to provide some estimates, within your manuscript, together with their associated references concerning the % of the cost reduction so that the reader can obtain a broad idea about the importance of improved scheduling?
Page 14, lines 641-642, “…Patients’ decisions may also be influenced by family members, ambulance crews,…”: Can you please explain how the decision of a patient is influenced by the crew of an ambulance?
Page 18, pages 896-897, “…there is a frequent shortage of nurses in the night shift – both during weekdays and weekends….”: Can you please clarify better within your manuscript which is the reason for this phenomenon?
Page 22, line 1077-1078, “…In the simulated experiments, we assume that the hospital has complete information about whether a patient will leave the unit within an hour or not…”: I am not sure about this assumption. I think that “one hour” is a rather small period.
Section 4. CONCLUSION: It is an excellent Section. However, I think that a general comment concerning “how far or how close we are to a situation where all these problems can be solved” will be helpful for the reader.
My suggestion is to place the references in numerical order within your manuscript. For example, reference [27], page 3, line 135 appears before references [3], [4] etc.
Reference List at the end of the paper: Please note that both names and surnames of the authors appear in reference [31] Heydari, Mohammad, Kin Keung Lai, and Zhou Xiaohu. Please correct accordingly and check all your references for such cases.
Page 29, APPENDIX B (TABLES), Table 1: Optimal staff schedules for a selected day-of-week: Please include “n.a.” (not available or not applicable) in the empty cells of the specific Table.
Please check your manuscript throughout for English grammar and syntax errors (e.g., Page 21, line 1039, First, we argue that if a hospital were….).
Author Response
Dear respected editorial board;
We thank you for the valuable suggestions and comments of the reviews. We have revised the paper accordingly. We are sure that the paper has improved its quality remarkably. Please find the attachment.

Reviewer 2 Report
The paper is well articulated. However, the following points are observed.
1. There are some grammatical issues for which the paper needs a thorough revision.
2. Equations (1) & (2) can be symbolized.
3. In (1), What does the dash (-) stands for? Is an OR connecting Supply and Demand? or it is an AND operator? clarity is sorted.
4. Kindly add a list of Nomenclature/Notations.
5. The future scope of the research is missing. Kindly add the potential extension of the work in the conclusion section.
6. The percentage of Older references is higher. Consider updating the paper with the most recent works (2021-2022) in the field.
Author Response

(The authors gave the same response as above.)
